# RUNX1, a transcription factor mutated in breast cancer, controls the fate of ER-positive mammary luminal cells

**Maaike PA van Bragt[1,2], Xin Hu[1,2,3†], Ying Xie[1,2†], Zhe Li[1,2]***

[1]Division of Genetics, Brigham and Women's Hospital, Boston, United States; [2]Department of Medicine, Harvard Medical School, Boston, United States; [3]School of Life Sciences, Jilin University, Changchun, China

**Abstract** *RUNX1* encodes a RUNX family transcription factor (TF) and was recently identified as a novel mutated gene in human luminal breast cancers. We found that *Runx1* is expressed in all subpopulations of murine mammary epithelial cells (MECs) except the secretory alveolar luminal cells. Conditional knockout of *Runx1* in MECs by *MMTV-Cre* led to a decrease in luminal MECs, largely due to a profound reduction in the estrogen receptor (ER)-positive mature luminal subpopulation, a phenotype that could be rescued by the loss of either *Trp53* or *Rb1*. Mechanistically RUNX1 represses *Elf5*, a master regulatory TF gene for alveolar cells, and regulates mature luminal TF/co-factor genes (e.g., *Foxa1* and *Cited1*) involved in the ER program. Collectively, our data identified a key regulator of the ER+ luminal lineage whose disruption may contribute to the development of ER+ luminal breast cancer when under the background of either *TP53* or *RB1* loss.

## Introduction

RUNX1, RUNX2, and RUNX3, and their common non-DNA-binding partner protein CBFβ, form a small family of heterodimeric transcription factors (TFs) referred to as Core-Binding Factors (CBFs) (*Speck and Gilliland, 2002*). They are best known as master regulators of cell fate determination in blood, bone, and neuron, respectively (*Chuang et al., 2013*). RUNX1 is a master regulator of hematopoietic stem cells and multiple mature blood lineages. Translocations and mutations involving both *RUNX1* and *CBFB* are frequently found in human leukemias (*Speck and Gilliland, 2002*). Recently, key roles of this family of TFs in epithelial cells and solid tumors also started to emerge (*Taniuchi et al., 2012*; *Chuang et al., 2013*; *Scheitz and Tumbar, 2013*). In particular, in breast cancer, recent whole-genome and whole-exome sequencing studies have consistently identified point mutations and deletions of *RUNX1* in human luminal breast cancers (*Banerji et al., 2012*; *Cancer Genome Atlas Network, 2012*; *Ellis et al., 2012*). In addition, mutations in *CBFB* were also identified in luminal breast cancers from these studies. Its gene product CBFβ is critical for enhancing DNA-binding by RUNX TFs through allosteric regulation (*Bravo et al., 2001*; *Tahirov et al., 2001*). Thus, we hypothesized that RUNX1, together with CBFβ, might play a key role in mammary epithelial cell (MEC) lineage determination as a master regulatory TF and that the loss of this normal function might contribute to breast cancer development.

There are two major epithelial cell lineages in the mammary gland (MG), luminal lineage (including ductal and alveolar luminal cells), and basal lineage (the mature cell type in the basal lineage is myoepithelial cell) (*Figure 1A*). These two types of MECs are produced by multipotent mammary stem cells (MaSCs, which are basal cells) during embryonic development or upon MEC transplantation to cleared mammary fat pads (*Shackleton et al., 2006*; *Stingl et al., 2006*; *Spike et al., 2012*). In adult MGs, they appear to be maintained by both lineage-specific unipotent stem cells and multipotent basal MaSCs, based on lineage tracing studies (*Van Keymeulen et al., 2011*; *van Amerongen et al., 2012*;

*For correspondence: zli4@rics.bwh.harvard.edu

†These authors contributed equally to this work

**Competing interests:** The authors declare that no competing interests exist.

**eLife digest** Stem cells can develop into the many types of specialized cell found in the body. Several proteins regulate these transformations by switching on and off the expression of genes that are specific to different cell types. Disrupting these proteins can cause the development of cells to go awry and can lead to cancer.

A protein called RUNX1 controls gene expression to direct the development of blood cells. Mutations in the gene encoding this protein have been linked to blood cancers and a particular type of breast cancer, which begins in the cells that line the ducts that carry milk towards the nipple.

Mammary duct-lining cells develop from a pool of stem cells that produces breast tissue cells. Now van Bragt et al. have found that RUNX1 is expressed in the cells lining the ducts of the mammary glands, except those that produce milk. Deleting the gene for RUNX1 in mice reduced the number of duct-lining cells, especially a subgroup of cells that are the sensors for the hormone estrogen. Through experiments on breast cancer cells, van Bragt et al. found that RUNX1 is able to dictate the fate of duct-lining breast cells by controlling other protein regulators. RUNX1 boosts the activity of at least one regulator that encourages the cells to become duct-lining cells and represses another regulatory protein that turns cells into milk-producing cells.

Next, van Bragt et al. found that, in mice lacking the gene for RUNX1, reducing the amounts of certain proteins that normally suppress the formation of tumors restored the populations of estrogen-sensing duct-lining cells. This suggests that mutations in the gene encoding RUNX1, coupled with the loss of a tumor-suppressing protein, may contribute to the development of cancer in the cells that line the breast ducts.

The next challenge is to determine exactly how RUNX1 mutations work together with the loss of the tumor-suppressing protein to drive breast cancer development. This knowledge may translate into new approaches to prevent or treat this type of breast cancer.

*Rios et al., 2014*; *Tao et al., 2014*; *Wang et al., 2014*). The gene regulatory network that must be in place to orchestrate lineage specification and differentiation of stem cells into mature MEC types remains largely elusive, although a number of key TFs have been identified in recent years, for example, GATA3 has been shown as a master regulator for both ductal and alveolar luminal cells (*Kouros-Mehr et al., 2006*; *Asselin-Labat et al., 2007*); ELF5 was identified as a master regulator of alveolar cells (*Oakes et al., 2008*; *Choi et al., 2009*); SLUG (SNAIL2) was shown as a master regulator of MaSCs, and it could reprogram differentiated MECs to transplantable MaSCs, together with another TF, SOX9 (*Guo et al., 2012*). In this work, we asked whether RUNX1 is an integral part of this transcription network and how its mutations contribute to breast tumorigenesis. By using genetic, cellular, and molecular approaches, we found that RUNX1 is a key regulator of estrogen receptor (ER)-positive mature ductal luminal cells, and that the loss of *RUNX1* may contribute to the development of ER+ luminal breast cancer when under the background of either *TP53* or *RB1* loss.

## Results

### *Runx1* is expressed in all MEC subsets except in alveolar luminal cells

We first measured expression levels of all three *Runx* genes and their common co-factor gene *Cbfb* in freshly sorted basal epithelial cells (Lin−CD24+CD29hi) and luminal epithelial cells (Lin−CD24+CD29lo) (*Figure 1A*) from adult virgin female mice by quantitative RT-PCR (qRT-PCR). Results showed that *Runx1* is the predominantly expressed *Runx* gene in both luminal and basal cells (*Figure 1B*). Immunohistochemical (IHC) staining further confirmed the expression of RUNX1 protein in these two major MEC types in adult virgin MGs (*Figure 1C*). However, RUNX1 expression is largely absent in alveolar luminal cells (ALs) that start to emerge during pregnancy (*Figure 1D–E*). In the lactating gland, the only MEC type that still expresses RUNX1 is the myoepithelial cell (*Figure 1F–G*). Upon involution, RUNX1 expression is restored to a pattern resembling that of the virgin gland (*Figure 1H*). Additionally, we performed microarray expression profiling of sorted subsets of MECs, including basal cells (Lin−CD24+CD29hi), luminal progenitors (LPs, Lin−CD24+CD29loCD61+), mature luminal cells (MLs, Lin−CD24+CD29loCD61−, mainly represent ductal luminal cells in virgin MGs), and alveolar luminal cells

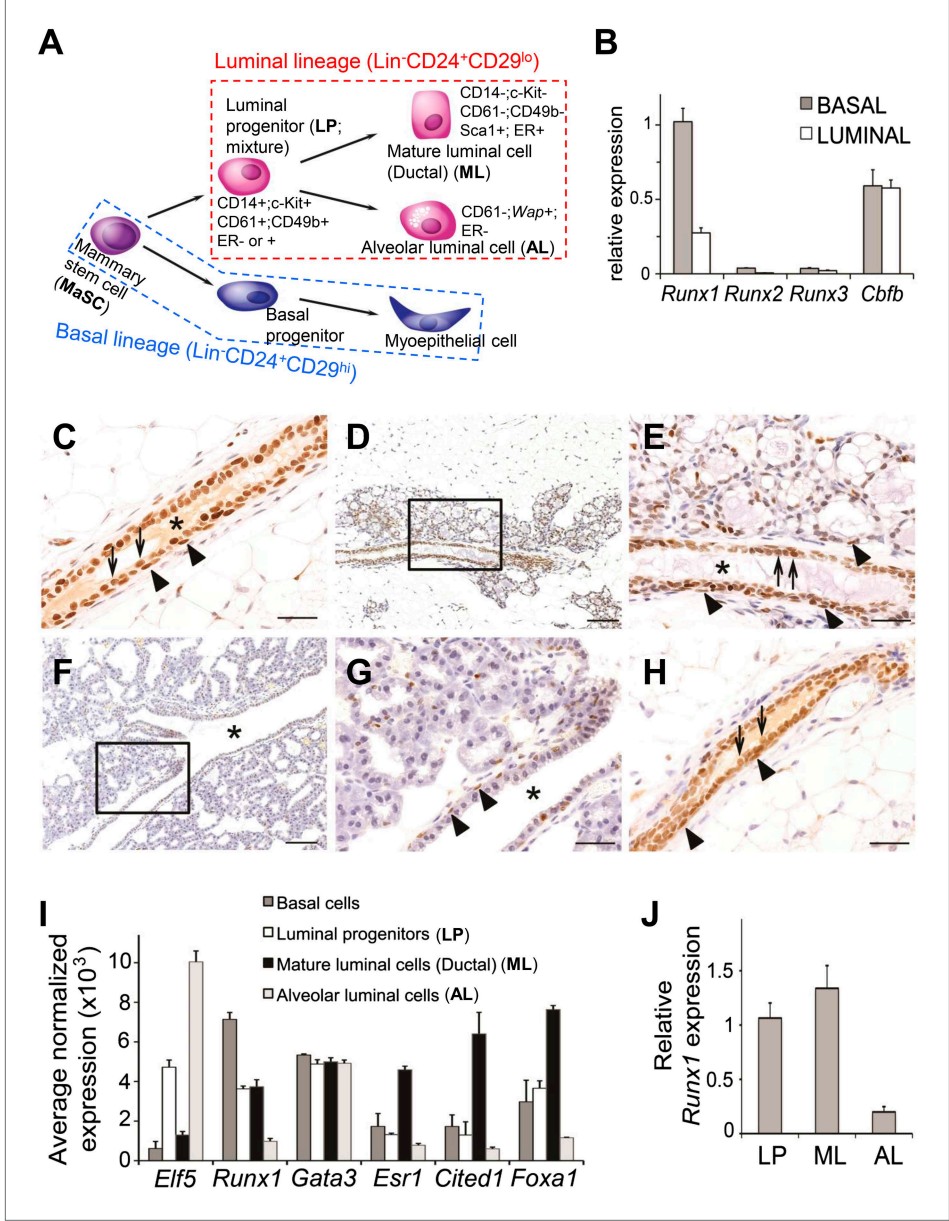

**Figure 1**. Expression pattern of *Runx1* in murine MGs. (**A**) Schematic diagram of a simplified version of the MEC hierarchy. MECs can be separated into the luminal and basal lineages. Major MEC subpopulations, their names and name abbreviations, as well as their marker expression patterns are shown. Note: 'luminal progenitor (LP)' has been used to refer to progenitor cells for the luminal lineage defined based on either CD61 (*Asselin-Labat et al., 2007*), or CD14 and c-Kit (*Asselin-Labat et al., 2011*), or CD49b (*Li et al., 2009*; *Shehata et al., 2012*), and is therefore a mixture of overlapping progenitor cell populations and may include common or separate progenitors for ductal and alveolar luminal cells. (**B**) qRT-PCR analysis of *Runx1, Runx2, Runx3*, and *Cbfb* transcripts isolated from luminal and basal cells of adult virgin female mice. (**C–H**) IHC staining for RUNX1 on sections of MGs at different developmental stages: (**C**) adult virgin, (**D–E**) mid-gestation (the region highlighted in **D** is shown in **E**), (**F–G**) lactation (the region highlighted in **F** is shown in **G**), and (**H**) after involution. Arrows and arrowheads indicate RUNX1-expressing luminal and basal cells, respectively; * indicates lumen. Scale bars = 20 µm. (**I**) Relative expression values of indicated genes determined by microarray analysis of the indicated MEC subpopulations isolated from the MGs of adult virgin female mice. ALs were isolated as YFP+ cells from *Wap-Cre;R26Y* females (i.e., MECs genetically marked by the *Wap-Cre* transgene) during mid-gestation. Affymetrix probes used to estimate expression of each indicated gene are 1419555_at, 1422864_at, 1448886_at, 1435663_at, 1449031_at, and 1418496_at for *Elf5*, *Runx1*,

*Figure 1. Continued on next page*

*Figure 1. Continued*

*Gata3*, *Esr1*, *Cited1*, and *Foxa1*, respectively. (**J**) *Runx1* expression levels were confirmed in sorted LPs, MLs, and ALs (as in **I**) by qRT-PCR.

The following figure supplement is available for figure 1:

**Figure supplement 1**. Expression analysis of *Runx1* and other select luminal transcription factor (TF) genes based on microarray.

(ALs, i.e., MECs genetically marked by *Wap-Cre* at mid-gestation; *Wap-Cre* is a transgenic mouse line with Cre expression under the control of the *Whey acidic protein* [*Wap*] promoter, a milk protein promoter [*Wagner et al., 1997*]). Estimation of *Runx1* levels based on this microarray dataset confirmed its expression in all MEC subsets except in ALs (*Figure 1I*). We examined *Runx1* expression levels in different subsets of MECs in several additional published microarray datasets (*Asselin-Labat et al., 2010*; *Lim et al., 2010*; *Meier-Abt et al., 2013*) and further confirmed this expression pattern (*Figure 1—figure supplement 1A–C*); in particular, in the pregnant MGs, *Runx1* was also found expressed in basal MECs but not in luminal MECs (mainly ALs) (*Figure 1—figure supplement 1C*). Lastly, by qRT-PCR, we verified that *Runx1* was indeed expressed in sorted LPs and MLs but not in *Wap-Cre*-marked ALs (*Figure 1J*).

## Loss of *Runx1* in MECs affects multiple MEC subsets

The *RUNX1* mutations identified from the recent sequencing studies of human breast cancers include point mutations, frame-shift mutations, and deletions (*Banerji et al., 2012*; *Cancer Genome Atlas Network, 2012*; *Ellis et al., 2012*). We analyzed the breast cancer-associated missense mutations of *RUNX1* to determine whether they lead to *loss-of-function* of *RUNX1* (*Figure 2A*). Based on a previous alanine-scanning site-directed mutagenesis study (*Li et al., 2003*), we found that these missense mutations either affect amino acid residues in RUNX1 that directly contact DNA, or disrupt the overall fold of its DNA-binding RUNT domain or abolish its binding to CBFβ, both of which would also perturb its DNA-binding (*Figure 2B*). Thus, similar to *RUNX1* deletions, the point mutations also lead to *loss-of-function* of *RUNX1*, due to disrupted DNA-binding ability. Therefore, we asked whether and how the loss of *Runx1* could affect the development of normal MECs.

*Runx1*$^{-/-}$ mice died during mid-gestation mainly due to hemorrhages in the central nervous system and are thus not suitable to determine the effect of *Runx1*-loss on MG development (*Okuda et al., 1996*; *Wang et al., 1996*). We therefore used a conditional knockout allele of *Runx1* (*Runx1*$^{L/L}$) (*Li et al., 2006b*). To facilitate characterization of *Runx1*-null MECs, we bred in a conditional Cre-reporter, *Rosa26-Stop-YFP* (*R26Y*). Cross of the floxed *Runx1* mice with the *R26Y* reporter mice and *MMTV-Cre* transgenic mice allowed us to simultaneously disrupt *Runx1* in MECs and mark the targeted cells by Yellow Fluorescent Protein (YFP) (*Figure 3A*). Lineage analysis revealed that in virgin MGs, *MMTV-Cre* mainly targeted MECs in the luminal lineage, but it could also lead to Cre-mediated recombination in a portion of basal MECs (*Figure 3B*). By fluorescence-activated cell sorting (FACS), we isolated YFP$^+$ MECs from *MMTV-Cre;Runx1*$^{L/L}$*;R26Y* females and *MMTV-Cre;Runx1*$^{L/+ and +/+}$*;R26Y* control females and by qRT-PCR, we confirmed the loss of *Runx1* expression in YFP$^+$ MECs from *MMTV-Cre;Runx1*$^{L/L}$*;R26Y* females (*Figure 3C*). Whole-mount analysis of MGs from *MMTV-Cre;Runx1*$^{L/L}$*;R26Y* virgin females or dams on lactation day-0 did not reveal any obvious gross morphological abnormalities, although a portion (3 out of 7) of *MMTV-Cre;Runx1*$^{L/L}$*;R26Y* females exhibited a slight delay in expansion of their ductal trees during pubertal growth (*Figure 3—figure supplement 1A*). Surprisingly, however, none of the *MMTV-Cre;Runx1*$^{L/L}$*;R26Y* dams were able to successfully nurse their pups (*Figure 3—figure supplement 1B*). Most of their pups died within 24 hr postpartum and no milk spots were observed in them compared to pups from *MMTV-Cre;Runx1*$^{+/+(or L/+)}$*;R26Y* dams. A closer examination of MGs of lactating *MMTV-Cre;Runx1*$^{L/L}$*;R26Y* females revealed milk stasis and an increasing number of cytoplasmic lipid droplets (*Figure 3—figure supplement 1C*). Similar phenotypes have also been observed in *Runx1* conditional knockout mice with *Krt14-Cre* (i.e., Cre-expressing transgenic mice under the control of the *Keratin 14* promoter) (*Krt14-Cre;Runx1*$^{L/L}$, data not shown) and in a number of genetically engineered mice with defects in myoepithelial cell contraction and milk ejection (*Li et al., 2006a*; *Plante et al., 2010*; *Haaksma et al., 2011*; *Weymouth et al., 2012*). Since *Runx1* is only expressed in myoepithelial cells at this stage (*Figure 1D–G,I*, and *Figure 1—figure supplement 1C*), we reasoned

## A

MASDSIFESFPSYPQCFMRECILGMNPSRDVHDASTSRRFTPPSTALSPGKMSEALPLGAPDAGAALAGK

LRSGDRSMVEVLADHPGELVRTDSPNFLCSVLPTHWRCNKTLPIAFKVVALGDVPDGTLVTVMAGNDENY

R166Q  G168E R169K

SAELRNATAAMKNQVARFNDLRFVGRSGRGKSFTLTITVFTNPPQVATYHRAIKITVDGPREPRRHRQKL

DDQTKPGSLSFSERLSELEQLRRTAMRVSPHHPAPTPNPRASLNHSTAFNPQPQSQMQDTRQIQPSPPWS

YDQSYQYLGSIASPSVHPATPISPGRASGMTTLSAELSSRLSTAPDLTAFSDPRQFPALPSISDPRMHYP

GAFTYSPTPVTSGIGIGMSAMGSATRYHTYLPPPYPGSSQAQGGPFQASSPSYHLYYGASAGSYQFSMVG

GERSPPRILPPCTNASTGSALLNPSLPNQSDVVEAEGSHSNSPTNMAPSARLEEAVWRPY

## B

| RUNX1 missense mutations based on Ellis et al | Additional missense mutations based on Taniuchi et al | Additional missense mutations based on TCGA, Nature 2012 | Affected amino acid residue based on Li et al | Mutation disrupts DNA contact | Mutation perturbs Runt domain fold | Mutation disrupts CBFβ contact |
|---|---|---|---|---|---|---|
| R166Q | | | R139 | Yes | | |
| G168E | | | G141 | | Yes | |
| R169K | | | R142 | Yes | | |
| | S73C | | S73 | | Yes | |
| | R174Q | | R174 | Yes | | |
| | | G108D | G108 | | | Yes |
| | | N112S | N112 | | Yes | |
| | | L134P | L134 (not studied) | Close to a critical DNA contact residue R135 | | |
| | | D171G | D171 | Yes | | |

**Figure 2**. Analysis of *RUNX1* mutations. *RUNX1* somatic missense mutations identified in human breast cancers disrupt its DNA-binding either directly (disrupting direct DNA contact) or indirectly (disrupting the overall protein fold of its DNA-binding RUNT domain or disrupting CBFβ binding). (**A**) RUNX1 full-length protein sequence; RUNT domain is highlighted in blue. The three amino acid residues affected by point mutations in luminal breast cancers (based on *Ellis et al. (2012)*) are shown in red. Several additional missense mutations (based on *Cancer Genome Atlas Network (2012)*; *Taniuchi et al. (2012)*) are also highlighted with red font. (**B**) How these missense mutations affect RUNX1 DNA-binding is predicted based on a previous structural and biochemical analysis of the RUNT domain (*Li et al., 2003*).

that the nursing defects observed are most likely due to a disrupted function of RUNX1 in myoepithelial cells. Additional studies are required to determine this.

In *MMTV-Cre;Runx1^{L/L};R26Y* virgin females, we found that the percentages of the YFP-marked MEC population (representing *Runx1*-null MECs) were significantly reduced when compared to those of the *MMTV-Cre;Runx1^{+/+};R26Y* control females (*Figure 3D–E*). Furthermore, the ratios of the YFP-marked luminal to basal subsets were also significantly reduced in *MMTV-Cre;Runx1^{L/L};R26Y* females (*Figure 3D,F*); this could be due to an expansion of the YFP-marked basal population or a reduction in the YFP-marked luminal population, or both. However, since the overall population of YFP⁺ MECs in

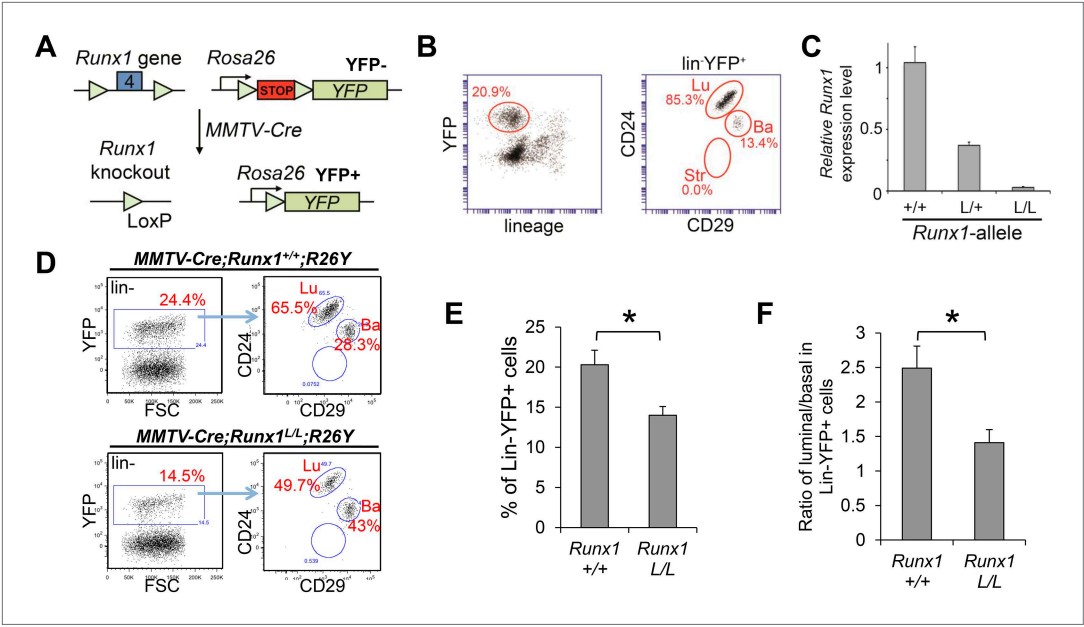

**Figure 3**. *Runx1*-loss leads to a reduction in the luminal MEC population. (**A**) Schematic representation of the *Runx1* conditional knockout allele in which its exon 4 is flanked by *lox*P sites, as well as the *R26Y* conditional Cre-reporter. STOP: transcriptional *stopper* cassette. Subsequent breeding with *MMTV-Cre* resulted in mice in which selected subsets of MECs express YFP and lack expression of functional RUNX1. (**B**) FACS gating strategy for detecting lin⁻YFP⁺ (lin: lineage markers) MECs, as well as YFP⁺ lin⁻CD24⁺CD29ˡᵒ luminal (Lu), and lin⁻CD24⁺CD29ʰⁱ basal (Ba) MECs in *MMTV-Cre;R26Y* females. Str: stromal cells. (**C**) qRT-PCR analysis confirming the loss of *Runx1* expression in YFP⁺ MECs sorted from *MMTV-Cre;Runx1^{L/L};R26Y* females (L/L). (**D**) FACS analysis showing the reduced lin⁻YFP⁺ MEC population (left plots), as well as the reduced lin⁻YFP⁺ luminal population (right plots), in *MMTV-Cre;Runx1^{L/L};R26Y* female compared to those in *MMTV-Cre;Runx1^{+/+};R26Y* control female. (**E**–**F**) The percentages of lin⁻YFP⁺ MEC population (**E**), as well as the ratios of luminal/basal subpopulations among the lin⁻YFP⁺ gate (**F**), are significantly reduced in *MMTV-Cre;Runx1^{L/L};R26Y* females (n = 9) (*L/L*) compared to those in *MMTV-Cre;Runx1^{+/+};R26Y* control females (n = 10) (*+/+*). p values: *: p ≤ 0.05; error bars represent mean ± S.E.M.

The following figure supplement is available for figure 3:

**Figure supplement 1**. Conditional knockout study of *Runx1* in murine MGs.

*MMTV-Cre;Runx1^{L/L};R26Y* females was reduced (*Figure 3D–E*), the reduction in the YFP⁺ luminal/basal ratio was most likely due to a reduction in the YFP-marked *Runx1*-null luminal population.

## Loss of *Runx1* leads to a profound reduction in ER⁺ mature luminal cells

Recent studies suggest that most breast cancers, including both basal-like and luminal subtypes, may originate from luminal cells, rather than from basal MaSCs (*Lim et al., 2009*; *Molyneux et al., 2010*; *Proia et al., 2011*; *Keller et al., 2012*). Furthermore, *RUNX1* and *CBFB* mutations have only been found in the luminal subtype of human breast cancers (*Banerji et al., 2012*; *Cancer Genome Atlas Network, 2012*; *Ellis et al., 2012*) and our data so far showed that the loss of *Runx1* appeared to lead to a reduction in the luminal population (*Figure 3D–F*), we therefore examined the role of RUNX1 in luminal MECs (from which luminal breast cancers may originate).

To determine the overall defects of *Runx1*-null luminal MECs, we first profiled the transcriptomes of YFP⁺ *Runx1*-null luminal cells (sorted from *MMTV-Cre;Runx1^{L/L};R26Y* females) and control YFP⁺ *Runx1*-wild-type (WT) luminal cells (sorted from *MMTV-Cre;Runx1^{+/+};R26Y*) by microarray. By gene set enrichment analysis (GSEA [*Subramanian et al., 2005*]), we observed significant enrichment of a previously generated LP signature and downregulation of a ML signature in *Runx1*-null luminal cells (*Figure 4A*). These LP and ML signatures were generated previously based on subset-specific genes conserved in the corresponding human and mouse MEC subpopulations (*Lim et al., 2010*). Furthermore, we also observed significant enrichment of multiple gene sets related to p53 signaling in *Runx1*-null luminal

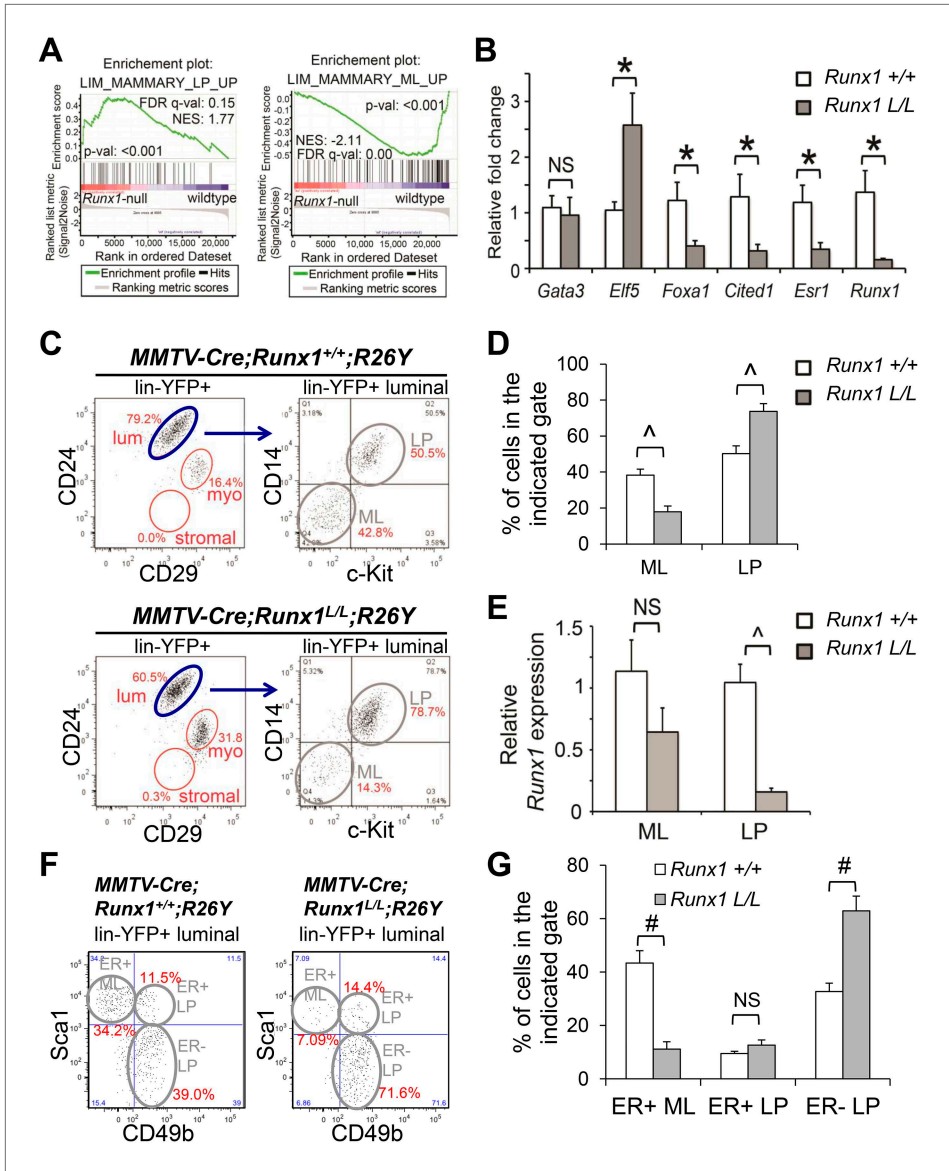

**Figure 4**. *Runx1* disruption leads to a profound reduction in ER[+] MLs. (**A**) GSEA enrichment plots showing correlation of the expression profiles of *Runx1*-null or WT luminal MECs with previously published conserved human and mouse signatures of LPs (left) or MLs (right) (***Lim et al., 2010***). (**B**) qRT-PCR validation of TF/co-factor genes known to play roles in luminal lineage specification and maintenance. RNA was isolated from YFP[+] *Runx1*-null and WT primary luminal MECs. (**C**) FACS plots of expression of CD14 and c-Kit, two LP markers (***Asselin-Labat et al., 2011***), in the gated YFP[+] luminal MECs (Lin[−]CD24[+]CD29[lo]) of adult *MMTV-Cre;Runx1[L/L];R26Y* virgin females and *MMTV-Cre;Runx1[+/+];R26Y* control females. Note the CD14[−]c-Kit[−] mature luminal (ML) subpopulation was largely lacking in the lower right plot. (**D**) Quantification of the percentages of the ML and LP subpopulations as indicated in **C**, showing significant reduction in the ML subpopulation in *MMTV-Cre;Runx1[L/L];R26Y* females (n = 13) compared to that in *MMTV-Cre;Runx1[+/+];R26Y* control females (n = 10). (**E**) qRT-PCR analysis showing significantly reduced *Runx1* expression in the LP subpopulation but not in the ML subpopulation in *MMTV-Cre;Runx1[L/L];R26Y* females. (**F**) FACS plots of expression of CD49b, a LP marker, and Sca1, an ER[+] ML marker (***Shehata et al., 2012***) in the gated YFP[+] luminal MEC population. Note the CD49b[−]Sca1[+] ER[+] ML subpopulation was dramatically reduced, whereas the CD49b[+]Sca1[−] ER[−] LP subpopulation was increased in *MMTV-Cre;Runx1[L/L];R26Y* females. (**G**) Quantification of the percentages of the ER[+] ML, ER[+] LP, and ER[−] LP subpopulations as indicated in **F**, showing significant reduction in the ER[+] ML subpopulation in *MMTV-Cre;Runx1[L/L];R26Y* females (n = 4) compared to those in *MMTV-Cre;Runx1[+/+];R26Y* control females (n = 4). p values: *: p ≤ 0.05; #: p ≤ 0.005; ^: p ≤ 0.0005; NS = not significant; error bars represent mean ± S.E.M.

*Figure 4. Continued on next page*

*Figure 4. Continued*

The following source data and figure supplements are available for figure 4:

**Source data 1**. (**A**) Gene sets from the MSigDB database C2-CGP (chemical and genetic perturbations, v3.1) enriched in *Runx1*-null luminal cells.
**Figure supplement 1**. Analysis of the luminal phenotype in *MMTV-Cre;Runx1^{L/L};R26Y* females.

cells in relation to *Runx1*-WT luminal cells (*Figure 4—figure supplement 1A*), possibly indicating a stress response in these mutant luminal cells in vivo. Lastly, we examined the expression levels of a number of TF/co-factor genes known to be part of the transcription network that regulates specification and maintenance of luminal MECs. In our microarray data, we found that *Elf5*, a TF gene critically required in the alveolar cell lineage and a LP marker (*Oakes et al., 2008*; *Choi et al., 2009*; *Lim et al., 2010*), was upregulated in *Runx1*-null luminal cells, whereas several ductal luminal TF/co-factor genes (e.g., *Gata3*, *Foxa1*, *Esr1*, *Cited1*) were downregulated (*Figure 4—figure supplement 1B*). Among these luminal TF/co-factor genes, *Foxa1* encodes a pioneer factor that is a key determinant of ERα (encoded by *Esr1*) function (*Bernardo et al., 2010*; *Hurtado et al., 2011*); *Cited1* encodes a selective co-activator for estrogen-dependent transcription, which potentially regulates the sensitivity of luminal cells to estrogen (*Yahata et al., 2001*; *Howlin et al., 2006*). When validated by qRT-PCR, we found that although the expression of *Gata3* did not seem to be significantly affected in luminal cells upon *Runx1*-loss, expression levels of *Foxa1*, *Esr1*, and *Cited1* were downregulated in *Runx1*-null luminal cells, whereas *Elf5* expression was upregulated (*Figure 4B*).

Our microarray data for the entire luminal population suggested that *Runx1*-loss in luminal MECs might lead to either a global block in luminal differentiation or loss of a mature luminal MEC subpopulation. To determine this, we performed FACS analysis of the Lin⁻CD24⁺CD29^lo luminal lineage. Intriguingly, we found that in both pubertal and adult virgin *MMTV-Cre;Runx1^{L/L};R26Y* females, the YFP⁺ *Runx1*-null ML subpopulation defined based on CD14 and c-Kit staining (Lin⁻CD24⁺CD29^loCD14⁻c-Kit⁻, referred to as CD14⁻c-Kit⁻ MLs hereafter) (*Asselin-Labat et al., 2011*) was significantly reduced, whereas the YFP⁺ *Runx1*-null LP subpopulation (Lin⁻CD24⁺CD29^loCD14⁺c-Kit⁺, referred to as CD14⁺c-Kit⁺ LPs hereafter) was increased (*Figure 4C–D*, *Figure 4—figure supplement 1C–D*). The reduction in the ML subpopulation was further confirmed in adult virgin *MMTV-Cre;Runx1^{L/L};R26Y* females based on CD61 staining (MLs: Lin⁻CD29^loCD61⁻) (*Asselin-Labat et al., 2007*) (*Figure 4—figure supplement 1E–F*).

The residual YFP⁺ MECs in the ML gate in *MMTV-Cre;Runx1^{L/L};R26Y* virgin females could either represent a truly *Runx1*-null ML subpopulation (but reduced in percentage) or represent YFP-marked MLs that have escaped Cre-mediated disruption of the *Runx1^L* allele (thus not truly *Runx1*-null). To determine this, we sorted YFP⁺ CD14⁻c-Kit⁻ MLs from *MMTV-Cre;Runx1^{L/L};R26Y* virgin females and *MMTV-Cre;Runx1^{+/+};R26Y* control females; as an internal control, we also sorted YFP⁺ CD14⁺c-Kit⁺ LPs from the same animals. By qRT-PCR, we found that while *Runx1* expression in YFP⁺ LPs from *MMTV-Cre;Runx1^{L/L};R26Y* females was significantly reduced, its expression in YFP-marked MLs was only slightly reduced (*Figure 4E*). This data suggested that many YFP⁺ MLs in *MMTV-Cre;Runx1^{L/L};R26Y* females might have escaped Cre-mediated excision in at least one copy of their *Runx1^L* alleles (thus they were either *Runx1^{+/+}* or *Runx1^{+/−}*). The data thus also suggests that RUNX1 is essential for the emergence or maintenance of the ML lineage.

A recent study demonstrated that the luminal cell compartment of the mouse MG could be further resolved into non-clonogenic ER⁺ MLs, as well as clonogenic ER⁺ LPs and ER⁻ LPs based on CD49b and Sca1 staining; the ER⁺ LPs may represent progenitors for ER⁺ MLs whereas the ER⁻ LPs probably represent alveolar progenitors (*Shehata et al., 2012*). We examined these luminal subpopulations in the YFP-gated luminal population in *MMTV-Cre;Runx1^{L/L};R26Y* virgin females. We found that compared to their corresponding subpopulations in *MMTV-Cre;Runx1^{+/+};R26Y* control females, the CD49b⁻Sca1⁺ ER⁺ ML subpopulation was significantly reduced in *MMTV-Cre;Runx1^{L/L};R26Y* females, whereas the CD49b⁺Sca1⁻ ER⁻ LP subpopulation was significantly increased; the CD49b⁺Sca1⁺ ER⁺ LP subpopulation was not significantly altered (*Figure 4F–G*). Of note, since the overall YFP⁺ luminal population was significantly reduced in *MMTV-Cre;Runx1^{L/L};R26Y* females (*Figure 3D–F*), the increase in the ER⁻ LP subpopulation might be mainly due to a reduction in the ER⁺ ML subpopulation (thus proportionally increased the percentage of the ER⁻ LP subset), rather than a significant expansion of ER⁻ LPs

per se; similarly, although the percentage of the ER+ LP subpopulation was not significantly changed, the absolute number of YFP+ ER+ LPs could still be reduced (due to an overall reduction in YFP+ luminal MECs). In support of this, we measured *Runx1* expression in these three luminal MEC subpopulations. We found that whereas the YFP-marked ER− LP subset had a profound reduction in *Runx1* expression, the YFP+ ER+ LP subset exhibited a partial reduction in *Runx1* transcripts, and the YFP+ ER+ ML subpopulation had almost no loss of *Runx1* expression (*Figure 4—figure supplement 1G*), suggesting RUNX1 is required for both ER+ LPs and ER+ MLs. Collectively, our data suggest that RUNX1 is required for the development or maintenance of the ER+ luminal lineage, and it is particularly essential for the ER+ MLs.

## Reduction in the ER+ luminal subpopulation upon *Runx1* disruption can be rescued by the loss of either *Trp53* or *Rb1*

From recent whole-genome/exome sequencing studies, *RUNX1* and *CBFB* mutations were only identified in the luminal subtype of human breast cancers (*Banerji et al., 2012*; *Cancer Genome Atlas Network, 2012*; *Ellis et al., 2012*), which are typically ER+. Paradoxically our data so far in the murine model suggest that *loss-of-function* of *Runx1* leads to a reduction in ER+ luminal MECs in vivo. Furthermore, we have followed *MMTV-Cre;Runx1^{L/L};R26Y* females for at least 18 months and have not observed any mammary tumor development in them. This can be explained by a possibility in which *RUNX1*-mutant breast cancer originates from ER+ luminal MECs and *Runx1* disruption alone actually leads to the loss of the cell-of-origin of such cancer. We hypothesized that additional genetic events might be needed to cooperate with *RUNX1*-loss to promote the development of luminal breast cancer from ER+ luminal MECs.

Interestingly, one recent sequencing study unveiled that pathway signatures of *RB1* mutation, *TP53* mutation, and *RUNX1* mutation are co-associated with human luminal B breast tumors (*Ellis et al., 2012*). Furthermore, by carefully examining luminal breast cancer cases with *RUNX1* mutations, we noticed that >50% of them are accompanied by mutations or deletions in either *TP53* or *RB1* genes (*Cancer Genome Atlas Network, 2012*). Lastly, our microarray data for luminal MECs suggested that loss of *Runx1* might lead to activation of the p53 pathway in luminal cells in general (*Figure 4—figure supplement 1A*). Based on these observations, we hypothesized that loss of *Runx1* in luminal MECs perturbs the fate of ER+ MLs, possibly leading to a stress response and subsequently upregulation of the p53 pathway, which then triggers cell cycle arrest (or apoptosis); this would cause the *Runx1*-null (YFP+) MLs to be outcompeted by *Runx1*-WT (YFP−) MLs. If this is the case, then either disruption of the p53 pathway or activation of cell cycle by *Rb1*-loss might rescue the phenotype of ER+ ML cell loss upon *Runx1* disruption. To test this, we bred *MMTV-Cre;Runx1^{L/L};R26Y* mice to *Trp53* or *Rb1* conditional knockout mice (*Trp53^{L/L}* or *Rb1^{L/L}*). In the resulting compound mice, we were only able to follow *MMTV-Cre;Runx1^{L/L};Trp53^{L/L};R26Y* or *MMTV-Cre;Runx1^{L/L};Rb1^{L/L};R26Y* females for ~4–5 months or ~9–10 months, respectively, due to lethality possibly caused by hematopoietic malignancies (as *MMTV-Cre* has leaky expression in bone marrow hematopoietic cells). Nevertheless, we were able to analyze MEC subpopulations in their MGs. Upon *MMTV-Cre*-induced *Trp53* or *Rb1* loss alone, the percentages of YFP-marked MECs increased dramatically so that the majority of MECs in their MGs became YFP+ (*Figure 5A*, increased from ~20–30% to ~70–90%), suggesting a growth advantage for *Trp53*-null or *Rb1*-null MECs (in relation to their *Trp53*-WT or *Rb1*-WT YFP− neighbors). However, the percentages of the YFP-marked luminal population were both reduced (*Figure 5A*, middle and bottom left plots compared to upper left plot, green circles). Interestingly, disruption of *Runx1* either together with *Trp53* or with *Rb1* significantly increased the percentages of the YFP+ luminal population (*Figure 5A–C*, increased from ~4% [*Trp53*-loss alone] to ~11% [*Runx1/Trp53*-loss] [*Figure 5B*] and from ~12% [*Rb1*-loss alone] to ~23% [*Runx1/Rb1*-loss] [*Figure 5C*], respectively). Of particular note, the percentage of the YFP-marked ML subpopulation, which was dramatically reduced upon *Runx1*-loss alone (*Figure 5A*, upper right plot, red circle), was reverted back to almost the normal level upon simultaneous loss of *Runx1* together with either *Trp53* or *Rb1* (*Figure 5A*, middle right plot for *Trp53* [*5B* for quantification], bottom right plot for *Rb1* [*5C* for quantification], red circles). To verify the presence of ER+ MECs in the MGs of these compound female mice, we performed IHC staining for ERα and could indeed detect abundant ERα+ luminal MECs in both *MMTV-Cre;Runx1^{L/L};Trp53^{L/L};R26Y* and *MMTV-Cre;Runx1^{L/L};Rb1^{L/L};R26Y* compound females (*Figure 5—figure supplement 1A–B*, since the majority of MECs in their MGs were YFP+, most of these ERα+ MECs should represent MECs with simultaneous loss of *Runx1* and *Trp53* or *Rb1*). As the residual YFP+ MECs in the ML gate from *MMTV-Cre;Runx1^{L/L};R26Y* females (*Runx1*-loss alone) appear to have escaped Cre-mediated excision in at least one *Runx1^L* allele (*Figure 4E*), we wanted to determine whether YFP+ MLs in these compound

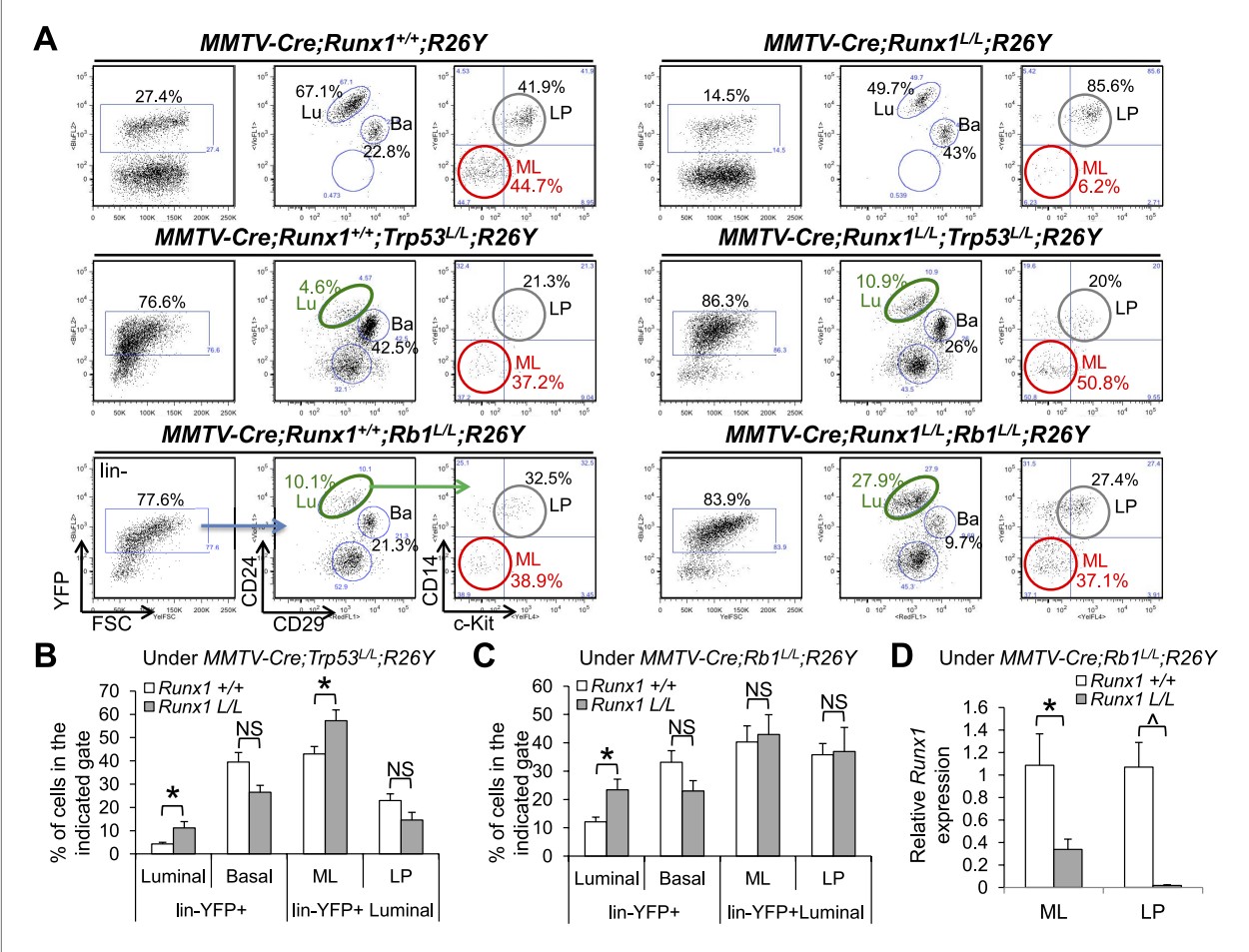

**Figure 5**. Reduction in ER$^+$ MLs upon *Runx1* disruption can be rescued by *Trp53* or *Rb1* loss. (**A**) FACS analysis showing total lin$^-$YFP$^+$ MEC population, lin$^-$YFP$^+$ luminal population, and lin$^-$YFP$^+$ ML and LP subpopulations (from left to right for each genotype, an example of the gating strategy is indicated in the bottom left plots) in female mice with the indicated genotypes. Those highlighted in green show increased lin$^-$YFP$^+$ luminal populations upon the loss of both *Runx1* and *Trp53* or *Rb1* (middle and bottom right plots, respectively) compared to those of *Trp53* or *Rb1* loss alone (middle and bottom left plots, respectively); those highlighted in red show increased lin$^-$YFP$^+$ ML subpopulations upon the loss of both *Runx1* and *Trp53* or *Rb1* (middle and bottom right plots, respectively) compared to that of *Runx1*-loss alone (upper right plot). Lu: luminal; Ba: basal; LP: luminal progenitor; ML: mature luminal cell. (**B–C**) Quantifications for the percentages of each indicated subpopulation in **A** under either the *Trp53*-loss (**B**) or *Rb1*-loss background (**C**); in (**B**) *Trp53*-loss alone (n = 5), *Trp53/Runx1*-loss (n = 3); in (**C**) *Rb1*-loss alone (n = 5), *Rb1/Runx1*-loss (n = 5). (**D**) qRT-PCR analysis showing significantly reduced *Runx1* expression in both the YFP-marked ML and LP subpopulations in *MMTV-Cre;Runx1$^{L/L}$;Rb1$^{L/L}$;R26Y* females. p values: *: p ≤ 0.05; ^: p ≤ 0.0005; NS = not significant; error bars represent mean ± S.E.M.

The following figure supplement is available for figure 5:

**Figure supplement 1**. Abundant ER$^+$ MECs are present in both *Runx1/Trp53*-null and *Runx1/Rb1*-null MGs.

mice had undergone (or escaped) Cre-mediated excision of their *Runx1$^L$* alleles. By qRT-PCR analysis, we observed more than 50% reduction in the *Runx1* expression level in the YFP-marked ML subpopulation sorted from *MMTV-Cre;Runx1$^{L/L}$;Rb1$^{L/L}$;R26Y* females (**Figure 5D**), suggesting a significant portion of these YFP$^+$ MLs should have undergone biallelic excision of their *Runx1$^L$* alleles.

## RUNX1 controls transcription of select target genes in vitro

Since *Runx1*-loss leads to a reduction in ER$^+$ MLs and the residual MECs present in the CD14$^-$c-Kit$^-$ or CD49b$^-$Sca1$^+$ ML gate appear to have escaped Cre-mediated disruption of *Runx1* (**Figure 4**, **Figure 4—figure supplement 1**), it is technically challenging to study how RUNX1 controls the fate of ER$^+$ luminal cells at the molecular level in this mouse model directly. Therefore, we first performed molecular studies in human breast cancer cell lines MCF7 and T47D. Although both cell lines are

ER+ luminal breast cancer cell lines, a key difference between them at the molecular level is that MCF7 cells express WT p53, whereas T47D cells carry a *TP53* missense mutation (nonfunctional p53) (*Schafer et al., 2000*). Interestingly, despite multiple attempts, we were only able to obtain *RUNX1* knockdown (kd) stable lines from *TP53*-mutant T47D cells but not from *TP53*-WT MCF7 cells. This observation suggests that a similar genetic interaction between *RUNX1*-loss and *TP53*-loss may also operate in human ER+ luminal breast cells. We therefore used T47D cells as our cell line model to study how RUNX1 controls the fate of ER+ luminal breast epithelial cells. By Western blot, we found that upon *RUNX1* kd, the protein level of ELF5 was increased, whereas the protein levels of both ERα and FOXA1 were reduced, and CITED1 protein level appeared unchanged (*Figure 6A*).

ELF5 is a master regulator of alveolar cells, a cell type in which *Runx1* is not expressed (*Figure 1D–J*, *Figure 1—figure supplement 1C*). Interestingly, it was shown previously that *RUNX1* is a direct target of ELF5 and is repressed by it, based on chromatin immunoprecipitation (ChIP) analysis

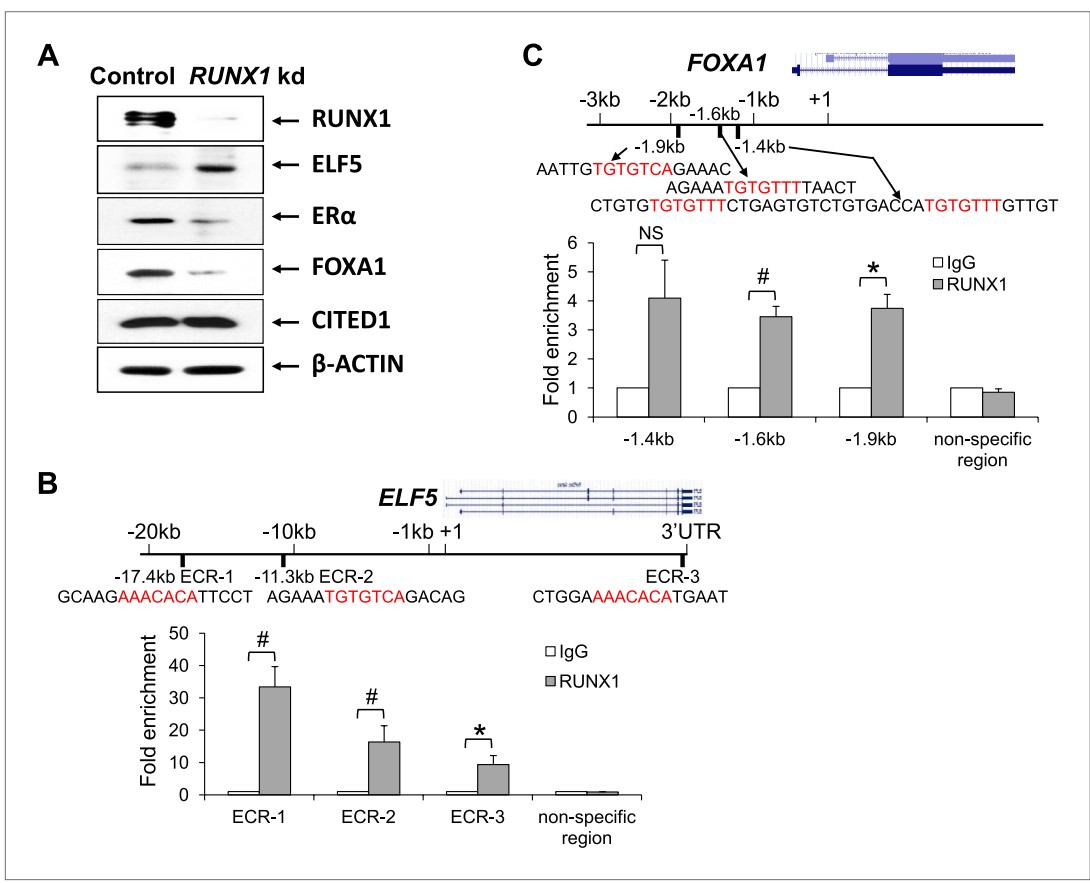

**Figure 6**. RUNX1 controls transcription of select target genes in human ER+ breast cancer cells. (**A**) Western blot showing upregulation of ELF5 and downregulation of ERα and FOXA1 upon RUNX1 knockdown (kd) in T47D luminal breast cancer cells. (**B**) ChIP analysis showing significant binding of RUNX1 to multiple ECRs (evolutionarily conserved regions) with RUNX-binding sites in the *ELF5* locus in T47D cells. (**C**) ChIP analysis showing significant binding of RUNX1 to the −1.6 kb and −1.9 kb regions of *FOXA1* in T47D cells. RUNX1-binding to the −1.4 kb region is marginally significant (p = 0.08). In (**B–C**), RUNX1-binding motifs (highlighted in red) and their flanking sequences are shown; note RUNX1-binding motifs in ECR-1 and ECR-3 of *ELF5* (**B**) are in the reverse strand. p values: *: p ≤ 0.05; #: p ≤ 0.005; NS = not significant; error bars represent mean ± S.E.M.

The following figure supplement is available for figure 6:

**Figure supplement 1**. Opposite expression patterns of RUNX1 and ELF5 proteins upon alveolar differentiation of HC11 cells.

(*Kalyuga et al., 2012*). In our microarray data for sorted MEC subsets, as well as those publicly available microarray datasets we analyzed, we could always observe a largely opposite expression pattern of *Elf5* and *Runx1* (*Figure 1I*, *Figure 1—figure supplement 1*). In both basal cells and MLs in which *Runx1* is highly expressed, *Elf5* is not; *Elf5* expression is greatly elevated in ALs whereas *Runx1* expression is repressed. This negative correlation in their expression levels could be further confirmed in the HC11 cell line model. While both *Runx1* and *Elf5* were expressed in uninduced HC11 cells, upon induction of alveolar differentiation, the ELF5 protein level was increased, whereas the RUNX1 protein level was reduced (*Figure 6—figure supplement 1*). To determine whether *ELF5* is also a direct target of RUNX1, we performed ChIP analysis on T47D cells and identified significant binding of RUNX1 to multiple evolutionary conserved RUNX-binding sites in the *ELF5* locus (*Figure 6B*). The RUNX1-binding was particularly profound in an enhancer region ~17 kb upstream of the *ELF5* transcription start site (ECR-1, *Figure 6B*).

Since *RUNX1* kd in T47D cells led to downregulation of ERα and FOXA1 (*Figure 6A*), we asked whether *ESR1* (encoding ERα) and *FOXA1* are direct targets of RUNX1. We identified a RUNX-binding motif in the *ESR1* control region ~1.4 kb upstream of its transcription start site, as well as several RUNX-binding motifs in the *FOXA1* control region ~1.4–1.9 kb upstream of its transcription start site. By ChIP assay, we confirmed significant binding of RUNX1 to the −1.6 kb and −1.9 kb motifs in the *FOXA1* locus (*Figure 6C*). Collectively, these data suggest that *FOXA1* and *ELF5* genes may be direct targets of RUNX1 positively and negatively regulated by it, respectively.

## RUNX1 represses *Elf5* and regulates mature luminal TF/co-factor genes involved in the ER program in vivo

To determine whether RUNX1 regulates the expression of these transcription regulators in primary cells in vivo, we took advantage of the rescue of *Runx1*-null ER⁺ luminal MECs by *Trp53* or *Rb1*-loss (*Figure 5*) and measured expression of these genes in FACS-sorted YFP⁺ luminal MEC subsets. We used *MMTV-Cre;Runx1^{L/L};Rb1^{L/L};R26Y* double mutant and *MMTV-Cre;Rb1^{L/L};R26Y* single mutant females for this analysis, as *MMTV-Cre;Runx1^{L/L};Trp53^{L/L};R26Y* double mutant females often exhibit early lethality. When comparing double mutants (with *Rb1/Runx1*-loss) to single mutants (with *Rb1*-loss alone), we found that both *Elf5* and *Esr1* appeared upregulated in ER⁺ LPs and ER⁺ MLs (based on CD49 and Sca1 staining) from double mutants (*Figure 7A–B*, left plots), and *Foxa1* and *Cited1* were downregulated in the rescued double mutant ER⁺ MLs (*Figure 7B*, left plot). Since we cannot rule out a possibility in which *Rb1*-loss in MECs also affects expression of these genes, we compared their expression in double mutants to matched WT females as well. From this comparison, we found that both *Elf5* and *Esr1* were also upregulated and *Foxa1* and *Cited1* were slightly downregulated in ER⁺ LPs and ER⁺ MLs from double mutants (*Figure 7A–B*, right plots). Furthermore, as we showed above, in *MMTV-Cre;Runx1^{L/L};R26Y* females, although the YFP-marked ER⁺ MLs appeared to have escaped Cre-mediated disruption of the *Runx1^L* allele, the YFP-marked ER⁺ LP subset did exhibit a partial reduction in *Runx1* expression (*Figure 4—figure supplement 1G*). We therefore asked whether there is any correlation of reduced *Runx1* expression to changes in expression of other genes in this MEC subset. Interestingly, we found that in *Runx1*-mutant ER⁺ LPs both *Elf5* and *Esr1* were upregulated and *Foxa1* and *Cited1* also appeared slightly upregulated (*Figure 7—figure supplement 1A*).

Thus, from both cell line and in vivo expression analyses, the gene that exhibits the most consistent change upon *Runx1*-loss is *Elf5*, which appears to be a target gene of RUNX1 repressed by it in ER⁺ luminal MECs. Intriguingly, we found that upregulation of *Elf5* upon *Runx1* disruption is not restricted to ER⁺ luminal cells and/or the *Rb1*-loss genetic background. In LPs defined based on either CD14⁺c-Kit⁺ or CD49b⁺Sca1⁻ where *Elf5* is abundantly expressed, loss of *Runx1* further increased their *Elf5* expression (*Figure 7—figure supplement 1B*). Strikingly, in basal MECs where *Elf5* is normally not expressed (*Figure 1I*, *Figure 1—figure supplement 1*), loss of *Runx1* led to its profound upregulation (*Figure 7—figure supplement 1B*). These data suggest that in normal MGs, RUNX1 represses expression of *Elf5* in almost all MEC subsets in which *Runx1* is expressed.

Our in vivo data showed that *Esr1* is upregulated rather than downregulated (based on the in vitro data in T47D cells, *Figure 6A*) in ER⁺ luminal MECs upon *Runx1*-loss. This is most likely due to hyperproliferation of *Runx1*-null ER⁺ luminal MECs under the *Rb1* (or *Trp53*)-null background, rather than de-repression of *Esr1* expression caused by *Runx1*-loss. Several lines of evidence support this notion. First, in *Runx1/Rb1*-double mutant females, we not only observed a slight increase in the percentages of total YFP⁺ MECs (*Figure 5A*) but also an increase in both the YFP⁺ luminal subset and, in particular,

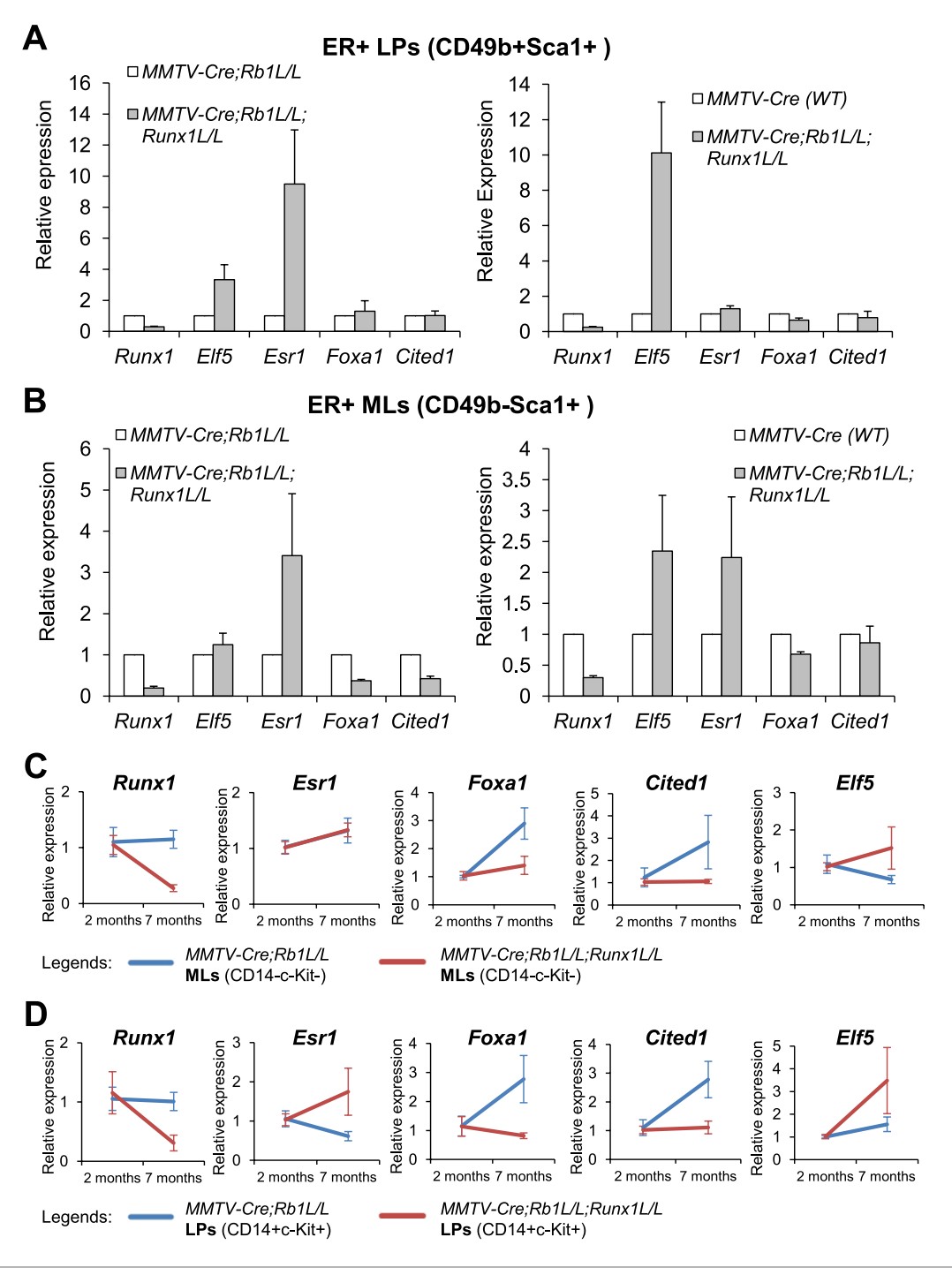

**Figure 7**. Target genes of RUNX1 in ER+ luminal MECs revealed by in vivo expression analysis. (**A–B**) qRT-PCR analysis showing changes in expression of the indicated genes in sorted YFP+ ER+ LPs (**A**) and ER+ MLs (**B**) (based on CD49b and Sca1 expression) from 4- to 5-month old *MMTV-Cre;Rb1L/L;Runx1L/L;R26Y* double mutant females, compared to those from either *MMTV-Cre;Rb1L/L;R26Y* single mutant females (left plots) or *MMTV-Cre;R26Y* WT females (right plots). (**C–D**) qRT-PCR analysis comparing expression of the indicated genes in CD14−c-Kit− MLs (**C**) and CD14+c-Kit+ LPs (**D**) from 7-month to 2-month old *MMTV-Cre;Rb1L/L;Runx1L/L;R26Y* double mutant and *MMTV-Cre;Rb1L/L;R26Y* single mutant (control) females. Expression was normalized to those of the corresponding 2-month old double or single mutant females, respectively. Error bars: mean ± S.E.M.

*Figure 7. Continued on next page*

*Figure 7. Continued*

The following figure supplements are available for figure 7:

**Figure supplement 1**. Loss of *Runx1* in vivo leads to changes in expression of ER program-related genes.
**Figure supplement 2**. RUNX1 reduction leads to hyperproliferation of abnormal ER+ luminal cells in a context-dependent manner.

the YFP-marked ER+ LP and ER+ ML subpopulations (*Figure 5A–C*, *Figure 7—figure supplement 2A*). In *TP53*-mutant T47D cells, we found that kd of *RUNX1* leads to a significant increase in their proliferation (*Figure 7—figure supplement 2B*). Furthermore, we quantified ERα+ luminal MECs in MGs with either *Runx1/Rb1*-loss or *Runx1/Trp53*-loss and found that both double mutants contained significantly more ERα+ MECs than single *Rb1*-loss or *Trp53*-loss mutants (*Figure 7—figure supplement 2C–D*). Since our expression analysis was based on FACS-sorted MEC subsets (e.g., LP, ML) and each subset may represent a mixture of both ER+ and ER– MECs (with different proportions), a change in this proportion, due to overrepresentation of the rescued ER+ luminal cells in the FACS-sorted MEC subpopulations from *Rb1/Runx1*-null double mutants, may contribute to the ostensibly higher *Esr1* transcript signals in double mutant ER+ LPs and ER+ MLs (*Figure 7A–B*). Lastly, we found that *Esr1* upregulation in vivo appears restricted to the *Runx1*-null ER+ luminal MECs; in ER– LPs and ER– basal MECs, we did not observe upregulation of *Esr1* expression upon *Runx1* disruption (*Figure 7—figure supplement 1C*). This is apparently different from negative regulation of *Elf5* by RUNX1, in which loss of *Runx1* leads to expression of *Elf5* even in basal cells in which *Elf5* is normally not expressed (*Figure 7—figure supplement 1B*). This is also different from a recent finding of repression of *Esr1* by ID4, as loss of *Id4* leads to widespread upregulation of *Esr1* expression in both luminal and basal MECs (*Best et al., 2014*). Collectively, these data suggest that *Esr1* is not a direct target repressed by RUNX1 in vivo; the downregulation of ERα in vitro in T47D cells upon *RUNX1* kd is likely to be indirect (e.g., due to *RUNX1* loss-induced upregulation of *ELF5*, as overexpression of *ELF5* in T47D cells can also suppress ERα expression [*Kalyuga et al., 2012*]).

The higher *Esr1* signal makes it challenging to accurately quantify any potential changes in expression levels of ER-related mature luminal genes upon *Runx1* disruption, by simply comparing their expression in double to single mutants or to WT controls (as the matched MEC subsets based on FACS sorting may have different cell compositions, if *Runx1*-null ER+ luminal MECs become over-populated). Interestingly, we noticed that in YFP+ MLs from *MMTV-Cre;Runx1^{L/L};Rb1^{L/L};R26Y* females, *Runx1* reduction became more profound in older females. We therefore similarly monitored changes in expression of other TF/co-factor genes over time in animals with the same genotype. This strategy may allow us to control for gene expression changes introduced by differences in cell populations or genetic backgrounds. By using this strategy, we found that in *Rb1*-null single mutants (controls), *Esr1*, *Foxa1*, and *Cited1* were upregulated and *Elf5* was downregulated in their YFP+ CD14–c-Kit– MLs when they aged; however, in YFP+ MLs from double mutants, following *Runx1* reduction, although *Esr1* was upregulated to a similar level (to that of single mutants), *Foxa1* and *Cited1* were not, and *Elf5* was even further upregulated (*Figure 7C*). We also observed a similar trend of changes for *Foxa1*, *Cited1*, and *Elf5* in YFP+ CD14+c-Kit+ LPs from the same animals (*Figure 7D*). Of note, *Esr1* expression in these LPs appeared further upregulated in older females, possibly due to hyperproliferation of the rescued ER+ luminal cells within this largely ER– LP subpopulation (the LP subset defined based on CD14+c-Kit+ contains a small number of ER+ cells [*Shehata et al., 2012*]). Overall, the data from this time course study further supports that in ER+ luminal cells, RUNX1 negatively and positively regulates the expression of *Elf5* and mature luminal TF/co-factor genes (e.g., *Foxa1* and *Cited1*), respectively.

As shown above, in YFP+ ER+ LPs from *MMTV-Cre;Runx1^{L/L};R26Y* females with partial *Runx1* reduction, we observed an abnormal expression pattern of these TF/co-factor genes (i.e., upregulation of both *Elf5* and *Esr1* and slight upregulation of *Foxa1* and *Cited1*, *Figure 7—figure supplement 1A*). This may be explained by a possibility in which a portion of them are committed for differentiation to ER+ MLs by upregulating *Esr1*; however due to *Runx1*-loss, *Elf5* is not repressed and *Foxa1* is not sufficiently upregulated in them, potentially leading to an abnormal population of *Elf5*+*Esr1*^{high}*Foxa1*^{low}*Cited1*^{low} ML-like cells retained in the ER+ LP FACS gate. A small number of such abnormal *Runx1*-null ER+ ML-like cells may also be present in the ML FACS gate. These abnormal luminal MECs may be the

target cells rescued for proliferation under the *Rb1*-loss background, and their hyperproliferation may contribute to the unusual $Elf5^+Esr1^{high}Foxa1^{low}Cited1^{low}$ expression pattern (*Figure 7C–D*). As a further support to this notion, we measured expression levels of these TF/co-factor genes in the ER⁻ LP, ER⁺ LP, and ER⁺ ML subsets sorted from WT animals and found that *Elf5* expression trends down, whereas expression of *Esr1*, *Foxa1*, and *Cited1* similarly trends up, from ER⁻ LPs to ER⁺ LPs and then to ER⁺ MLs, and that *Runx1* expression is also elevated from ER⁺ LPs to ER⁺ MLs (*Figure 7—figure supplement 2E*). This expression pattern suggests that differentiation of ER⁺ luminal MECs requires coordinated expression of these factors and *Runx1*-loss may disrupt their coordinated expression.

Collectively, our in vitro and in vivo expression analyses coupled with ChIP analysis suggest that *Elf5* is a key target gene of RUNX1 repressed by it in MECs. RUNX1 also positively regulates the expression of mature luminal TF/co-factor genes involved in the ER program and among them, *Foxa1* is a direct target of RUNX1, and RUNX1 does not appear to regulate transcription of *Esr1* directly.

## Discussion

Among TFs that control cell fates of the two subpopulations of luminal MECs, GATA3 has been shown as a common master regulator for both ER⁺ ductal luminal cells and ER⁻ alveolar luminal cells (*Kouros-Mehr et al., 2006*; *Asselin-Labat et al., 2007*), whereas ELF5 has been identified as a key regulatory TF specific for the alveolar luminal subset (*Oakes et al., 2008*; *Choi et al., 2009*). However, what is the TF that specifically controls the fate of the ER⁺ ductal luminal subset remained largely elusive. In this study, we identified RUNX1 as a key regulator of ER⁺ luminal MECs. RUNX1 controls the in vivo fate of this luminal subpopulation by repressing the program for an alternative cell fate choice (i.e., repressing the key TF gene for alveolar cells, *Elf5*) and by optimizing activation of the ML gene expression program (i.e., regulating key mature luminal TF/co-factor genes such as *Foxa1*) (*Figure 8A*). Loss of *Runx1* impairs the fate of ER⁺ luminal cells, leading to a profound reduction in this luminal subpopulation. However, the loss of either *Trp53* or *Rb1* can rescue this defect, leading to hyperproliferation of *Runx1*-mutant ER⁺ luminal cells, which may eventually progress to ER⁺ luminal breast cancer, upon acquisition of additional mutations (*Figure 8B*). Our study thus provides a direct link between a somatically mutated lineage-specific TF, impaired cell fate, and development of luminal breast cancer.

Among RUNX1 target genes, the repressed *Elf5* is of particular interest, as it encodes a master regulatory TF for the alternative cell fate of the milk-secreting alveolar lineage in which *Runx1* is not expressed (*Figure 1D–G,I–J*). We showed that *ELF5* is a direct target of RUNX1 and is repressed by it (*Figures 6A–B and 7*). Thus, combined with the previous observation in which *RUNX1* was reciprocally shown as a direct target repressed by ELF5 (*Kalyuga et al., 2012*), these data suggest that RUNX1 and ELF5 are two master regulators for mutually exclusive cell fate choices (i.e., ductal vs alveolar fates) by antagonizing each other's transcription program (e.g., RUNX1 promotes the ER program [this study], whereas ELF5 suppresses it [*Kalyuga et al., 2012*]) (*Figure 8A*) in a way similar to the GATA1-PU.1 paradigm for regulating the choice between erythroid and myeloid fates (*Huang et al., 2007*).

Intriguingly, RUNX1 not only represses *Elf5* expression in ER⁺ luminal cells but also in all other MEC subsets in which *Runx1* is expressed (*Figure 7—figure supplement 1B*). The de-repression of *Elf5* in basal MECs may also be of clinical relevance. Recently it was found that RUNX1 protein expression correlates with poor prognosis in ER⁻ breast cancer and more specifically in triple-negative breast cancer (TNBC) (*Ferrari et al., 2014*). Furthermore, *RUNX1* was also found associated with super-enhancers in an ER⁻ breast cancer cell line (*Hnisz et al., 2013*). As super-enhancers often associate with key oncogenes in cancer cells (*Loven et al., 2013*), these recent findings suggest that RUNX1 may also play an oncogenic role in ER⁻ breast cancers. The link between *RUNX1* and *ELF5* in basal MECs may explain a potential oncogenic role of RUNX1 in ER⁻ breast cancer/TNBC, as it was shown previously that *SNAI2* (encodes SLUG) is a target of ELF5 repressed by it (*Chakrabarti et al., 2012*). Thus, it is possible that RUNX1 expression in ER⁻ breast cancer cells may repress *ELF5* expression, leading to de-repression (thus upregulation) of *SNAI2* expression, which then promotes epithelial-mesenchymal transition (EMT) and aggressiveness of breast cancer cells. Interestingly, it was shown recently that *Snai2*-null mice exhibit a nursing defect, due to failed milk ejection caused by defects in basal/myoepithelial cell differentiation (*Phillips et al., 2014*). In *Runx1*-null mice, upregulation of *Elf5* in basal MECs may lead to repression of *Snai2*, which may provide an explanation for the similar nursing defect we have observed in our *Runx1* conditional knockout mice (*Figure 3—figure supplement 1B–C*).

In luminal breast cancer, our study provides strong evidence to support that RUNX1 plays a key role in this breast cancer subtype as a tumor suppressor in ER⁺ ductal luminal cells, which may be their cells

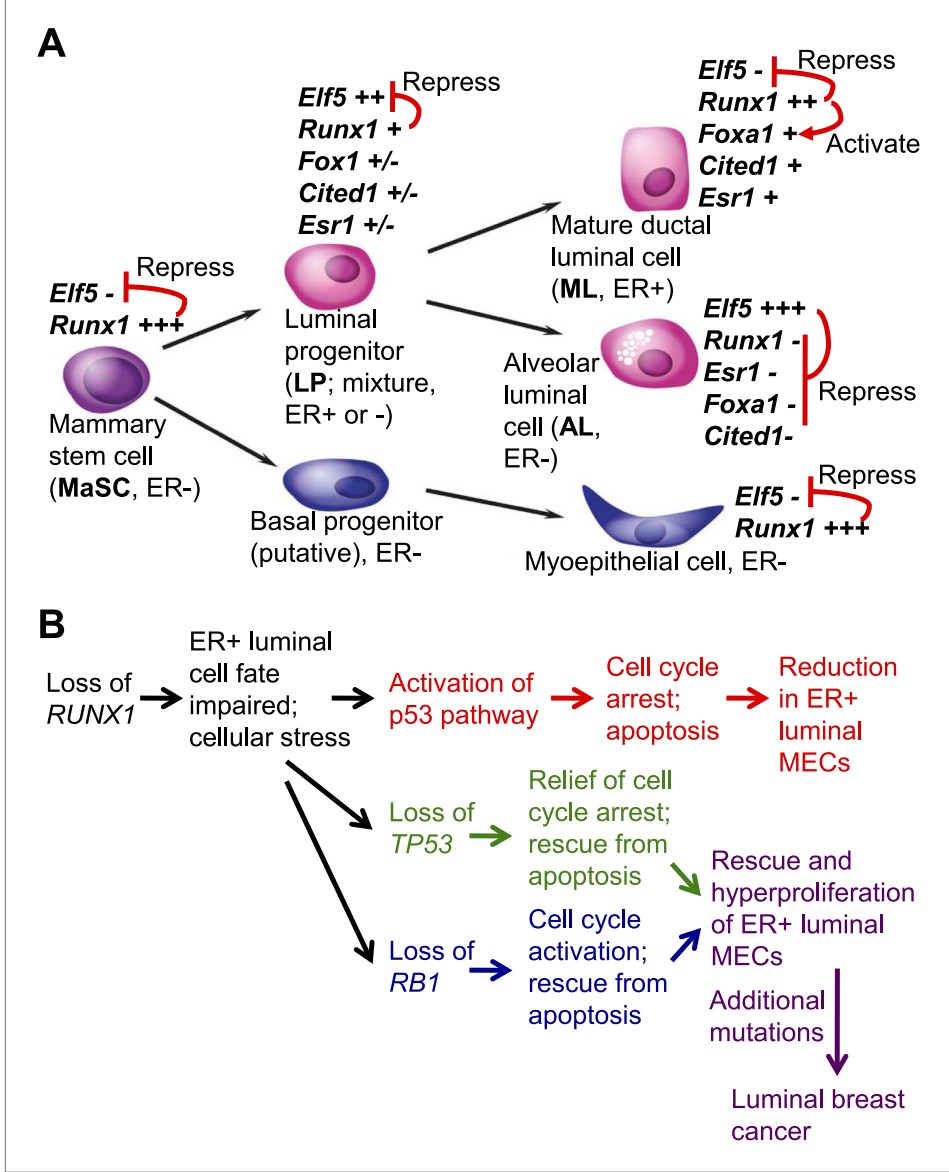

**Figure 8**. Model for the role of RUNX1 in ER+ mammary luminal cells and luminal breast cancer. (**A**) Relative expression levels of key TFs in different subsets of MECs are indicated ('+++', '++', '+', '±', '−' indicate highest to low to no expression, based on [**Figure 1I**] and our single cell profiling data for sorted MECs [MPAvB and ZL, unpublished data]). RUNX1 or ELF5 controls the ductal or alveolar luminal cell fate, respectively, by antagonizing each other. RUNX1 further controls the fate of ER+ ductal luminal MECs by regulating the ER program via modulating FOXA1 expression. (**B**) Genetic interaction between the loss of *RUNX1* and the loss of either *TP53* or *RB1* plays a key role in the development of *RUNX1*-mutant ER+ luminal breast cancer.

of origin. All three RUNX TFs have been shown to play context-dependent roles in breast cancer development as either tumor suppressors or oncogenes (*Chimge and Frenkel, 2013*). Among them, RUNX3 is also a tumor suppressor as it is often inactivated in human breast cancers and loss of one copy of *Runx3* led to spontaneous mammary tumor development in a portion of aged female mice (*Huang et al., 2012*). The tumor suppressor role of RUNX3 in breast cancer is explained by its ability to inhibit ERα-dependent transactivation by reducing the stability of ERα (*Huang et al., 2012*). In contrast, RUNX2 mainly exhibits oncogenic roles in breast cancer by promoting invasiveness and metastasis via its target, *SNAI2* (*Chimge et al., 2011*); however it may also play a tumor suppressor role in breast cancer by antagonizing ERα (thus, similar to RUNX3) (*Chimge et al., 2012*). In this study, we

showed that RUNX1, the most abundantly expressed RUNX TF in MECs, controls the fate of ER$^+$ luminal cells in part by upregulating *FOXA1* and repressing *ELF5*. Furthermore, RUNX1 has also been shown as a novel tethering factor for recruiting ERα to its genomic sites for ER-mediating transcriptional activation (*Stender et al., 2010*). Estrogen signaling has dual roles in MECs and breast cancer cells; on one hand it has an oncogenic role by promoting proliferation of ER$^+$ luminal breast cancer cells, on the other hand it also has a tumor suppressor role by promoting MEC differentiation and inhibiting metastasis of breast cancer cells (*Chimge and Frenkel, 2013*). The tumor suppressor role of RUNX2 and RUNX3 mainly relates to the antagonism between RUNX2/3 and the cancer-promoting program of ER signaling, whereas the tumor suppressor role of RUNX1 largely correlates to its ability to positively regulate the tumor-suppression program of ER signaling. The tumor suppressor role of RUNX1 is also consistent with a previous observation in which *RUNX1* was found among a 17-gene signature associated with metastasis as a gene downregulated in metastasis-prone solid tumors, including breast cancer (*Ramaswamy et al., 2003*).

Lastly, our study also provides an explanation for the paradox in which RUNX1 is a positive regulator of the ER program, yet its *loss-of-function* mutations and deletions are only present in ER$^+$ human luminal breast cancers (often accompanied by mutations or copy number losses in *TP53* or *RB1* genes) (*Cancer Genome Atlas Network, 2012*; *Ellis et al., 2012*). We show that the loss of *Runx1* does not appear to affect transcription of *Esr1* directly (thus, the affected luminal cells remain phenotypically ER$^+$) but may lead to a crippled ER program, in part due to de-repression of *Elf5* and insufficient upregulation of *Foxa1*, which may reduce the sensitivity and output of ER signaling, respectively (*Hurtado et al., 2011*; *Kalyuga et al., 2012*). The impaired ER program in *Runx1*-mutant ER$^+$ luminal cells may cause cellular stress, leading to activation of the p53 pathway and subsequently cell cycle arrest and/or apoptosis; as a result, abnormally differentiated *Runx1*-mutant ER$^+$ luminal cells are outcompeted by their WT neighbors in vivo. However, the loss of *Trp53* or *Rb1* can relieve the cell cycle arrest or positively activate cell cycle, respectively, and/or rescue apoptosis in them, leading to rescue of the *Runx1*-mutant ER$^+$ luminal cells. In humans, upon acquisition of additional mutations, the *RUNX1/TP53*-mutant or *RUNX1/RB1*-mutant ER$^+$ premalignant luminal cells may progress to ER$^+$ luminal breast cancer, upon acquisition of additional oncogenic events (*Figure 8B*). Of note, germline mutations of *RUNX1* that result in haploinsufficiency of RUNX1 can lead to an autosomal dominant disorder referred to as familial platelet disorder with a propensity to acute myeloid leukemia (FPD/AML) (*Song et al., 1999*). Interestingly, in one study that characterized three FPD/AML pedigrees, it was found that one female patient with FPD/AML also developed a breast cancer 2 years after AML was diagnosed, and no other tumors were observed in all three pedigrees (*Preudhomme et al., 2009*). Although the sample size for this study was too small, it certainly raises an intriguing question as to whether germline mutations of *RUNX1* predispose FPD/AML patients to luminal breast cancer, but only under a background of either *TP53* or *RB1* loss.

In summary, we identified RUNX1 as a key regulator of the ER$^+$ luminal lineage. Loss of *RUNX1* may contribute to the development of ER$^+$ luminal breast cancer under a background of either *TP53* or *RB1* loss and upon cooperation with other additional oncogenic events.

## Materials and methods

### Mice

Mice carrying the floxed *Runx1* allele (*Runx1$^{L/L}$*) (*Li et al., 2006b*) were bred with mice carrying a conditional Cre-reporter, *R26Y*. Subsequently, these mice were bred with mice that drive expression of Cre recombinase under the control of the mouse mammary tumor virus (*MMTV*) promoter (*MMTV-Cre*) and with mice carrying the floxed *Trp53* allele (*Trp53$^{L/L}$*) or floxed *Rb1* allele (*Rb1$^{L/L}$*). For studying *Runx1* disruption in basal MECs, Cre transgenic mice under the control of the *Keratin 14* promoter (*Krt14-Cre*) were also used. Mice were obtained from JAX (*R26Y*: 006148; *MMTV-Cre*: 003553) or the MMHCC repository (*Krt14-Cre*: 01XF1; *Wap-Cre*: 01XA8) or were a generous gift from Dr Stuart Orkin (*Trp53$^{L/L}$* and *Rb1$^{L/L}$* [*Walkley et al., 2008*]). All animal experiments and procedures were approved by our Institutional Animal Care and Use Committee (IACUC).

### Whole-mount, histology, and immunohistochemistry

Whole-mounts of MGs of pubertal, adult virgin, or lactation day-0 mice were fixed and processed as previously described (*Jones et al., 1996*). For histology and immunohistochemical staining, MGs were

fixed in 10% formalin and embedded in paraffin. For RUNX1 or ERα detection, antigen retrieval (Citrate buffer pH 6.0, 20 min boil in microwave oven) was performed prior to incubation with an anti-RUNX1 antibody (2593-1, Epitomics, Burlingame, CA) or an anti-ERα antibody (SC-542, Santa Cruz Biotechnology, Dallas, TX). Signal was detected using the impress reagent kit and DAB substrate (MP-7401 and SK-4100, Vector Laboratories, Burlingame, CA).

## Mammary gland cell preparation, flow cytometric analysis, and cell sorting

Thoracic and inguinal mammary glands were dissected from pubertal or adult virgin female mice and cell suspensions were prepared as previously described (*Shackleton et al., 2006*). Flow cytometric analysis was performed with a DXP11 analyzer (Cytek, Fremont, CA) or an Accuri C6 analyzer (BD Biosciences, San Jose, CA). FACS sorting was performed with a FACSAria sorter (BD Biosciences). Data were analyzed with FlowJo (Tree Star, Ashland, OR) or CFlow (BD Biosciences). Antibodies used for FACS were purchased from eBiosciences (San Diego, Ca) and included CD24-eFluor450, CD24-eFluor605, CD29-APC, CD61-PE, c-Kit-PE-CY7, CD14-PE, CD49b-PE, Sca1-APC and biotinylated CD31, CD45, and TER119 (i.e., lineage [Lin] markers), as well as Streptavidin-PerCP-CY5.5. We also used a Sca1-APC-CY7 antibody purchased from BD biosciences (San Jose, CA).

## Microarray analysis and quantitative RT-PCR

Total RNA from sorted subsets of MECs was prepared by the RNeasy kit (Qiagen, Valencia, CA) and amplified with the Ovation RNA Amplification System V2 (Nugen, San Carlos, CA). YFP-marked luminal cells were sorted from adult virgin *MMTV-Cre;Runx1^{L/L};R26Y* or *MMTV-Cre;Runx1^{+/+};R26Y* littermates. Normal MEC subsets, including MaSCs, LPs, and MLs, were sorted from WT C57/B6 adult virgin females; alveolar luminal cells (ALs) were sorted as YFP+ cells from *Wap-Cre;R26Y* females at mid-gestation. Mouse Genome 430 2.0 Array (Affymetrix, Santa Clara, CA) was used to generate the expression profiles. All arrays were normalized by dCHIP and analyzed by GSEA as described (*Subramanian et al., 2005*), using MSigDB database v3.1 (http://www.broadinstitute.org/gsea/msigdb/index.jsp). For qRT-PCR, cDNA was generated with Omniscript (Qiagen) according to the manufacture's protocol and real-time PCR was performed using FastStart SYBR Green Master (Roche, Indianapolis, IN). ΔΔCt method was used for normalization to the control group and to the endogenous control (*Hprt*). Primers are listed in *Supplementary file 1*.

## ChIP analysis

Cells were cross-linked with 1% formaldehyde at room temperature for 10 min, quenched with 0.125 M glycine for 5 min and washed with PBS, harvested by scraping and lysed in cell lysis buffer (0.1% SDS; 0.5% NP40; 1 mM EDTA; 10 mM Tris–HCl, pH 7.4; 0.5% NaDOC). 200–1000 bp DNA fragments were obtained after sonication. After 10 min centrifugation at max speed at 4°C, supernatant was used for IP overnight at 4°C. 30 μl Dynabeads Protein G beads (Invitrogen, Carlsbad, CA) and 1 μg antibody were used for each IP. One tenth of lysate was saved as input. The following antibodies were used: rabbit anti-RUNX1 (ab92336, Abcam, Cambridge, MA), rabbit IgG (sc-2027, Santa Cruz Biotechnology). The beads were washed twice with the following buffers, 3 min each: low-salt buffer (0.1% SDS; 1% Triton X-100; 1 mM EDTA; 10 mM Tris–HCl, pH 7.4; 300 mM NaCl; 0.1% NaDOC), high-salt buffer (0.1% SDS; 1% Triton X-100; 1 mM EDTA; 10 mM Tris–HCl, pH 7.4; 500 mM NaCl; 0.1% NaDOC), LiCl buffer (10 mM Tris–HCl, pH 8; 0.25M LiCl; 1 mM EDTA, pH 8; 1% NP-40; 1% NaDOC), and TE. Precipitated materials were eluted with 300 μl elution buffer (1% SDS; 0.1 M NaHCO3; 50 mM Tris–HCl, pH 8; and 10 mM EDTA). Chromatin was reverse-cross-linked by adding 12 μl of 5 M NaCl and incubated overnight at 65°C. DNA was obtained after RNaseA treatment, protease K treatment, phenol/chloroform extraction, and ethanol precipitation. DNA was analyzed by qPCR, normalized to the input DNA. Primers are listed in *Supplementary file 1*.

## *RUNX1* knockdown, Western blot, and proliferation assay

shRNAs for *RUNX1* were purchased from Open Biosystems (Huntsville, AL; shRNA sequences are listed in *Supplementary file 1*, data from a pool of TRCN0000013659-D1 and TRCN0000013662-D4 were shown). After lentiviral infection and puromycin selection, stable shRNA-expressing cell lines were generated. For Western blotting, whole-cell extracts were prepared by boiling cells for 10 min at 95°C in SDS sample buffer (50 mM Tris [pH 6.8]; 100 mM DTT; 2% SDS; 0.1% bromophenol blue; 10% glycerol). Cell lysates were then resolved by SDS-PAGE. β-actin (Fisher Lab, Hampton, NH) was

used as a loading control. Primary antibodies (RUNX1: Abcam ab92336, ELF5: Abcam ab77007, CITED1: Abcam ab92550, ERα: Santa Cruz Biotechnology sc-8002, FOXA1: Santa Cruz Biotechnology sc-6553) were detected using HRP-conjugated anti-rabbit antibodies and visualized using enhanced chemiluminescence detection (ECL reagents from Fisher Lab). Proliferation of T47D cells was determined by absorbance of alamarBlue, following manufacturer's protocol (Invitrogen Lot155363SA). $1 \times 10^5$ cells were seeded in a 96-well plate and were measured after 3 or 5 days in culture. 1/10 volume of alamarBlue reagent was directly added to cells in culture medium, incubated for 4 hr at 37°C. Absorbance of alamarBlue was monitored at 570 nm, using 600 nm as a reference wavelength (normalized to the 600 nm value).

## Statistical analysis

The results were reported as mean ± S.E.M. unless otherwise indicated, and Student's $t$ tests were used to calculate statistical significance.

## Accession numbers

The microarray expression profiling datasets generated in this manuscript have been deposited to the GEO database under the following accession numbers: GSE47375 (for Runx1) and GSE47376 (for normal MEC subsets) or as SuperSeries GSE47377.

## Acknowledgements

We are grateful to Y Qiu and G Losyev for expert technical assistance with FACS sorting, Y Shao and E Fox for microarray, R Bronson for histology, A Cantor and H Huang for providing $Runx1^{L/L}$ mice, S Orkin and A Chen for providing $Trp53^{L/L}$ and $Rb1^{L/L}$ mice, and L Ellisen for providing HC11 cells. This research was supported by a Pathway-to-Independence K99/R00 grant from NCI (CA126980), a Seed Grant from Harvard Stem Cell Institute, Milton Fund Award from Harvard University, Hearst Foundation Young Investigator Award from Brigham and Women's Hospital (BWH), and Start-up Fund from BWH to ZL; ZL is also supported by NIH R01 HL107663.

## Additional information

### Funding

| Funder | Grant reference number | Author |
|---|---|---|
| National Institutes of Health | K99/R00 CA126980, R01 HL107663 | Zhe Li |
| Harvard University | Harvard Stem Cell Institute Seed Grant SG-0062-10-01 | Zhe Li |
| Harvard University | Milton Fund Award | Zhe Li |
| Brigham and Women's Hospital | Hearst Foundation Young Investigator Award | Zhe Li |
| Brigham and Women's Hospital | Start-up fund | Zhe Li |

The funders had no role in study design, data collection and interpretation, or the decision to submit the work for publication.

### Author contributions

MPAB, Conception and design, Acquisition of data, Analysis and interpretation of data, Drafting or revising the article, Contributed unpublished essential data or reagents; XH, YX, Acquisition of data, Analysis and interpretation of data; ZL, Conception and design, Analysis and interpretation of data, Drafting or revising the article, Contributed unpublished essential data or reagents

### Ethics

Animal experimentation: This study was performed in strict accordance with the recommendations in the Guide for the Care and Use of Laboratory Animals of the National Institutes of Health. All of the animals were handled according to approved institutional animal care and use committee (IACUC) of Boston Children's Hospital (where the animals are housed) under protocol # 11-10-2034.

# Additional files

## Supplementary file

• Supplementary file 1. Primers and shRNAs used in this study (all sequences from 5′ to 3′).

## Major dataset

The following datasets were generated:

| Author(s) | Year | Dataset title | Dataset ID and/or URL | Database, license, and accessibility information |
|---|---|---|---|---|
| van Bragt MP, Li Z | 2013 | Microarray expression profiling study of Runx1-null and wild type luminal mammary epithelial cells | GSE47375; https://www.ncbi.nlm.nih.gov/geo/query/acc.cgi?acc=GSE47375 | In the public domain at GEO: http://www.ncbi.nlm.nih.gov/geo/. |
| van Bragt MP, Li Z | 2013 | Microarray expression profiling of distinct subsets of mouse mammary epithelial cells | GSE47376; https://www.ncbi.nlm.nih.gov/geo/query/acc.cgi?acc=GSE47376 | In the public domain at GEO: http://www.ncbi.nlm.nih.gov/geo/. |

The following previously published datasets were used:

| Author(s) | Year | Dataset title | Dataset ID and/or URL | Database, license, and accessibility information |
|---|---|---|---|---|
| Meier-Abt F, Thiry S, Roloff T, Bentires-Alj M | 2013 | Early parity-induced gene expression in mouse mammary cell subtypes | GSE40875; http://www.ncbi.nlm.nih.gov/geo/query/acc.cgi?acc=GSE40875 | In the public domain at GEO: http://www.ncbi.nlm.nih.gov/geo/. |
| Smyth GK, Wu D, Lim E, Visvader JE, Lindeman GJ | 2010 | Gene expression profiles of normal mouse mammary cell subpopulations FACS sorted based on expression of biomarkers | GSE19446; http://www.ncbi.nlm.nih.gov/geo/query/acc.cgi?acc=GSE19446 | In the public domain at GEO: http://www.ncbi.nlm.nih.gov/geo/. |
| Smyth GK, Wu D, Asselin-Labat M, Lindeman GJ, Visvader JE | 2010 | Mouse mammary epithelial cell subpopulations from pregnant and virgin mice | GSE20402; http://www.ncbi.nlm.nih.gov/geo/query/acc.cgi?acc=GSE20402 | In the public domain at GEO: http://www.ncbi.nlm.nih.gov/geo/. |

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
