## [Decision Letter]

Thank you for sending your work entitled “RUNX1, a transcription factor mutated
in breast cancer, controls the fate of ER-positive mammary luminal cells” for
consideration at *eLife*. Your article has been favorably evaluated by
Fiona Watt (Senior editor), a Reviewing editor, and 3 reviewers.

The Reviewing editor and the other reviewers discussed their comments before we reached
this decision, and the Reviewing editor has assembled the following comments to help you
prepare a revised submission.

This manuscript was submitted over a year ago, and after full review it was rejected by
*eLife*. In that paper, the expression and role of Runx1 in mammary
gland development and lineage formation was analyzed, and it was shown that Runx1 is
expressed in most of the mammary lineages, and its knockout via MMTV-Cre and K14-Cre
affects lactation. The authors showed that the number of mature ductal luminal cells
(ML) is decreased while their progenitors upstream (luminal progenitor LP) are increased
in Runx1 null relative to wild type (WT) controls. The authors concluded that Runx1 must
play a role in mature luminal ductal cell formation and that its loss blocks
differentiation. From there, the authors tried to understand the mechanisms of Runx1
action and found that in luminal cells the cells associated with LP are increased and
the genes associated with ML are decreased. Moreover they showed by ChIP in a cancer
cell line (MCF7) that Runx1 is bound to the promoters of Cited 1 (a ML gene) and Elf5 (a
LP gene). These genes have been previously studied and the phenotypes described seem to
match with the role uncovered here for Runx1, suggesting that Runx1 is directly
targeting these genes *in vivo*.

The reviewers felt that in the first submission, the paper brought novel insight into
the developmental role of Runx1 in breast biology, and felt that the association with
breast cancer was intriguing. However, they also felt that the paper required
significant editing and delineated specific suggestions to revise it. Additionally, the
reviewers felt that neither the developmental nor the cancer side of the paper were
sufficiently strong and necessitated experimental bolstering. The reviewers required
more experiments to bolster these stories, including providing more statistical analyses
and details of the methodology.

Two of the initial reviewers felt that the paper has changed significantly in the last
year, and that the revised version is now much better focused. That said, none of the
reviewers felt that the revised version is without problems. Since the paper was an
initial rejection, and since the revised work is now quite different from the initial
version, the reviewers and managing editor made the decision to treat this as a new
submission.

The reviewers agree that since the only ML YFP+ cells detected are ones that still
express Runx1, the authors are able to conclude that Runx1 is needed intrinsically for
the formation of the ER+ ML lineage. The reviewers were also impressed by the
authors' discovery that they discovered that the mutations found in human ER+
breast cancer are actually null mutations, as they liked their sleuthing that suggests
that the cells most likely (not unequivocal) the origin of these types of cancer
mutations require Runx1 in some way. The fact that the authors were able to rescue the
phenotype with RbKO and p53KO makes a compelling link with cancer. These findings were
deemed the strongest and a reasonable case for further consideration by
*eLife*.

That said, the reviewers remain unconvinced by the transcriptional mechanisms provided,
and do not feel that the authors have provided a compelling case for Runx1 being a
master regulator of the luminal lineage. Although carrying out ChIP-seq analyses on
*in vivo* material seems beyond the scope of current technology for
the mammary epithelial field, RNA-seq or microarray on purified populations of
*in vivo* material still seems within the grasp of what can be
reasonable to request. The authors argue that they can't do this because the
residual YFP-marked MECs in the ML gate are not truly Runx1-null MLs. And the population
of LPs is a mixed population of progenitors for either alveolar luminal or ductal
luminal cells. Therefore, they don't have evidence that the YFP-marked LPs are
progenitors upstream of MLs in the luminal differentiation hierarchy and are blocked in
differentiation to ER+ MLs. I agree that this poses a hurdle but it still does not
address the caveat that at present the data are restricted to culture studies with a
cell line. The authors are encouraged to explore other possible Cre lines in an effort
to bolster the *in vivo* expression data. The inducible KO line does not
have to be lineage-specific if coupled with cell surface markers to be used soon after
the induction. The authors should expect to see the genes downregulated in the LP cells
before the numbers of ML cells actually change. The data as they presently stand could
be misleading.

If the authors fail to rectify this point, one alternative would be to remove or shorten
and tone down the mechanistic data on how Runx1 regulates other transcription, and then
look a bit more closely on how RB/p53 and Runx1 interact. Another possibility would be
to look in the p53null Runx1nullYFP+ cells and explore their defects, if any. If
these cells appear as normal ML cells, this would suggest that the Runx1 null defect is
in proliferation/survival of the ML precursors. RB and p53 may not be directly related
to the differentiation phenotype and the targets shown, but rather to survival with loss
of function mutants. Even with only a small fraction of ML cells being knocked out
efficiently in the p53null Runx1null or RBnull Runx1 null double mutant, if the authors
can see Runx1 down in this mixed fraction, then they should be able to see other genes
changed that came up in their *in vitro* data. They could at least try
qRT-PCR, instead of microarray, on the genes they think are essential from *in
vitro* data, and then see whether these genes are changed in the double
mutant. This would strengthen their conclusions.

The authors either need to molecularly link this to differentiation or overcome the
hurdles for bolstering the *in vitro* data with *in vivo*
gene expression data. It would seem that one of these two avenues might be successful,
which is needed to clear the path to publishing in *eLife*.

---

## [Author Response]

*The reviewers remain unconvinced by the transcriptional mechanisms provided, and
do not feel that the authors have provided a compelling case for Runx1 being a master
regulator of the luminal lineage. Although carrying out ChIP-seq analyses on in vivo
material seems beyond the scope of current technology for the mammary epithelial
field, RNA-seq or microarray on purified populations of in vivo material still seems
within the grasp of what can be reasonable to request. The authors argue that they
can't do this because the residual YFP-marked MECs in the ML gate are not truly
Runx1-null MLs. And the population of LPs is a mixed population of progenitors for
either alveolar luminal or ductal luminal cells. Therefore, they don't have
evidence that the YFP-marked LPs are progenitors upstream of MLs in the luminal
differentiation hierarchy and are blocked in differentiation to ER+ MLs. I agree
that this poses a hurdle but it still does not address the caveat that at present the
data are restricted to culture studies with a cell line. The authors are encouraged
to explore other possible Cre lines in an effort to bolster the in vivo expression
data. The inducible KO line does not have to be lineage-specific if coupled with cell
surface markers to be used soon after the induction. The authors should expect to see
the genes downregulated in the LP cells before the numbers of ML cells actually
change. The data as they presently stand could be misleading*.

We agree that an inducible Cre line [e.g., *K8-CreER* for luminal mammary
epithelial cells (MECs)] combined with the *Runx1* conditional knockout
allele (*Runx1*^*L*^) and a conditional
Cre-reporter [e.g., *Rosa26-Stop-YFP (R26Y)*] may allow us to
characterize early molecular changes upon induced *Runx1* disruption in
ER^+^ luminal MECs. However, this would require extensive mouse
breeding in order to put multiple alleles together in the same mouse; therefore it was
not feasible for us to use this approach to obtain new *in vivo* data
within two months or an even longer time window. We therefore decided to still focus on
our existing *MMTV-Cre*-based mouse models and performed extensive
*in vivo* expression analysis using sorted MEC subsets from these
animals.

*If the authors fail to rectify this point, one alternative would be to remove or
shorten and tone down the mechanistic data on how Runx1 regulates other
transcription, and then look a bit more closely on how RB/p53 and Runx1 interact.
Another possibility would be to look in the p53null Runx1nullYFP+ cells and
explore their defects, if any. If these cells appear as normal ML cells, this would
suggest that the Runx1 null defect is in proliferation/survival of the ML precursors.
RB and p53 may not be directly related to the differentiation phenotype and the
targets shown, but rather to survival with loss of function mutants. Even with only a
small fraction of ML cells being knocked out efficiently in the p53null Runx1null or
RBnull Runx1 null double mutant, if the authors can see Runx1 down in this mixed
fraction, then they should be able to see other genes changed that came up in their
in vitro data. They could at least try qRT-PCR, instead of microarray, on the genes
they think are essential from in vitro data, and then see whether these genes are
changed in the double mutant. This would strengthen their conclusions*.

We have mainly followed this suggestion to obtain our *in vivo*
expression data for genes that have already been tested in our initial cell culture
model. Our main strategy was to monitor levels of *Runx1* reduction in
different subsets of MECs sorted from our *MMTV-Cre*-based conditional
knockout mice and then to determine whether there was any correlation of changes in
expression of other genes to reduced *Runx1* expression. Took advantage
of the rescue of *Runx1*-null ER^+^ luminal MECs by
Rb1-loss, we performed expression analysis in the rescued ER^+^ LPs and
ER^+^ MLs from the *Rb1/Runx1*-null mice (we did not
perform this experiment in *p53/Runx1*-null mice as they exhibited early
lethality, which prohibited us from obtaining enough animals for expression analysis).
In addition, we also performed expression analysis for these genes in the
ER^+^ LP subset from the *Runx1* conditional knockout
mice
(*MMTV-Cre;Runx1*^*L/L*^*;R26Y*),
as ER^+^ LPs may represent the precursor population for
ER^+^ MLs and our new *Runx1* expression analysis data
shows that this population has partial *Runx1* reduction (Figure 4—figure supplement 1). These
*in vivo* expression analysis data are presented in several new
figures, including Figure 7, Figure 7—figure supplement 1 and Figure 7—figure supplement 2, and are described in a new
sub-section in the main text.

Below is a summary of key findings from our new expression analysis experiments:

A) Consistent with our *in vitro* data, we also obtained strong
*in vivo* data to demonstrate that *Elf5* is a key
target gene repressed by RUNX1. In addition to its repression by RUNX1 in
ER^+^ luminal MECs, we found *Elf5* is repressed by
RUNX1 in almost all other MEC subsets (including basal MECs) in which
*Runx1* is expressed. The de-repressed expression of
*Elf5* in basal cells also allows us to provide an explanation for the
nursing defects observed in our *Runx1* conditional knockout mice (via a
potential *Runx1-Elf5-Snail2* link in basal/myoepithelial cells). Since
the main focus of this paper is to determine the role of RUNX1 in ER^+^
luminal MECs and luminal breast cancer, we did not pursue this further (but this
certainly opens a new avenue for a future study).

B) Unexpectedly, from our *in vivo* expression analysis, we found that
*Esr1* is upregulated in ER^+^ luminal MECs. This is
very different from our previous *in vitro* data in T47D cells in which
we observed a reduction in ERα level upon *RUNX1* knockdown.
Although we showed previously that RUNX1 binds to a RUNX1-binding motif ∼1.4kb
upstream of the *ESR1* transcription start site, this binding is
relatively weak (∼3-fold enrichment). We repeated the ChIP analysis for this
site, as well as for all the other sites we tested previously for *ELF5*,
*FOXA1* and *CITED1*. By applying statistical analysis,
we found that while RUNX1 binding to the control regions of *ELF5*,
*FOXA1* and *CITED1* is all statistically significant,
its binding to the -1.4kb site of *ESR1* is not
significant. Although we cannot rule out RUNX1 binding to other sites in the
*ESR1* locus, all these data suggest that RUNX1 may not regulate
transcription of *ESR1* directly. The downregulation of ERα in
T47D cells upon *RUNX1* knockdown is most likely indirect [e.g., due to
*RUNX1*-loss-induced upregulation of *ELF5*, as
overexpression of *ELF5* in T47D cells can also suppress ER expression
(25)]. To explain why
*Esr1* appears upregulated *in vivo* upon
*Runx1* reduction, we discussed several possibilities and in
particular, we provided evidence to support that the observed *Esr1*
upregulation may be in part due to hyperproliferation of the rescued
*Runx1*-null ER^+^ luminal MECs (i.e., more
*Esr1*-expressing cells present in a sorted MEC subpopulation compared
to the same subpopulation from control mice).

C) We provided evidence to support that *Foxa1* and
*Cited1* are target genes of RUNX1. *Runx1*-loss
reduces their expression levels but does not abolish their expression entirely. Their
downregulation is more profound in the rescued *Runx1*-null
ER^+^ MLs (new Figure 7), the subpopulation of luminal MECs that is affected
by *Runx1*-loss the most. To strengthen this conclusion, we used multiple
normalization approaches (i.e., normalized to *Rb1*-null single mutant
control mice, to wild type control mice, and to younger mice with the same genotype) to
control for gene expression changes introduced by differences in cell populations and/or
genetic backgrounds.

*The authors either need to molecularly link this to differentiation or overcome
the hurdles for bolstering the in vitro data with in vivo gene expression data. It
would seem that one of these two avenues might be successful, which is needed to
clear the path to publishing in* eLife*.*

Our new *in vivo* expression data, combined with our previous
observations, allow us to propose the following revised model: In the
ER^+^ luminal lineage, RUNX1 is mainly required during differentiation
from ER^+^ LPs to ER^+^ MLs (this is also the stage during
which *Runx1* expression is notably elevated, Figure 7—figure supplement 2, new data). Loss of
*Runx1* leads to abnormally differentiated ER^+^ MLs
(i.e.,
*Elf5*^+^*Esr1*^high^*Foxa1*^low^*Cited1*^low^
ML-like cells, due to de-repression of *Elf5* and insufficient
upregulation of *Foxa1* and *Cited1*, all of which may
impair the sensitivity and/or output of the ER program). This molecular defect may cause
cellular stress and subsequently activation of the p53 pathway in these abnormal cells,
leading to cell cycle arrest and/or apoptosis; as a result, abnormally differentiated
*Runx1*-null ER^+^ luminal cells are outcompeted by
their wild type neighbors *in vivo*. However, loss of p53 or Rb1 would
relieve the cell cycle arrest or positively activate cell cycle, respectively, and/or
rescue apoptosis in them, leading to rescue of these abnormal ER^+^
ML-like cells. Upon acquisition of additional mutations, these
*Runx1/p53*-mutant or *Runx1/Rb1*-mutant
ER^+^ premalignant luminal cells eventually progress to
ER^+^ luminal breast tumors. We believe this revised model has provided
important *in vivo* mechanistic advances to better explain why and how
*loss-of-function* of *RUNX1* leads to development of
ER^+^ luminal breast cancer specifically.